# PALEO-PGEM v1.0: A statistical emulator of Pliocene-Pleistocene climate

Philip B. Holden[1], Neil R. Edwards[1], Thiago F. Rangel[2], Elisa B. Pereira[2], Giang T. Tran[3] and Richard D. Wilkinson[4]

[1]Earth, Environment and Ecosystems, The Open University, Walton Hall, Milton Keynes MK7 6AA, UK
[2]Departmento de Ecologia, Universidade Federal de Goiás, CP 131, 74.001-970 Goiânia, Goiás, Brazil
[3]GEOMAR Helmholtz Centre for Ocean Research Kiel, Düsternbrooker Weg 20, 24105 Kiel, Germany
[4]School of Mathematics and Statistics, University of Sheffield, Sheffield, UK

*Correspondence to*: Phil Holden (Philip.holden@open.ac.uk)

**Abstract.** We describe the development of the "Paleoclimate PLASIM-GENIE emulator" PALEO-PGEM and its application to derive a downscaled high-resolution spatiotemporal description of the climate of the last five million years. The 5-million-year time frame is interesting for a range of paleo-environmental questions, not least because it encompasses the evolution of humans. However, the choice of time-frame was primarily pragmatic; tectonic changes can be neglected to first order, so that it is reasonable to consider climate forcing restricted to the Earth's orbital configuration, ice-sheet state and the concentration of atmosphere $CO_2$. The approach uses the Gaussian process emulation of the singular value decomposition of ensembles of the intermediate complexity atmosphere-ocean GCM PLASIM-GENIE. Spatial fields of bioclimatic variables of surface air temperature (warmest and coolest seasons) and precipitation (wettest and driest seasons) are emulated at 1,000 year intervals, driven by time-series of scalar boundary-condition forcing ($CO_2$, orbit and ice-volume), and assuming the climate is in quasi-equilibrium. Paleoclimate anomalies at climate model resolution are interpolated onto the observed modern climatology to produce a high-resolution spatiotemporal paleoclimate reconstruction of the Pliocene-Pleistocene.

## 1 Introduction

A high-resolution climate reconstruction of the Pliocene-Pleistocene will provide an unprecedented opportunity to advance understanding of many long-standing hypotheses about the origin and maintenance of biodiversity. Climate is among the strongest drivers of biodiversity and has played an important role throughout the history of life on Earth (Svenning et al 2015). Indeed, changes in climate over time have influenced core biological patterns and processes such as diversification, adaptation, species distribution and ecosystem functioning (Svenning et al 2015, Nogués-Bravo et al 2018). However, studies on the relationship between climate and biodiversity are still limited by the lack of high-resolution deep-time spatiotemporal paleoclimatic estimates, as the few studies available are at very sparse time slices (Lima-Ribeiro et al 2015). Thus, a high-resolution spatiotemporal paleoclimate data series of the past 5 million years will be useful to address many pressing questions on biodiversity dynamics. For instance, did the onset of glacial cycles promote more extinctions than recent climate cycles? Do species hold "evolutionary memory" of the warmer temperature of the Miocene? How did biodiversity respond to the increase in strength and frequency of glacial cycles during the Pliocene? Such knowledge is essential to understand biodiversity patterns and to forecast how organisms will respond to the current anthropogenic climatic change (Nogués-Bravo et al 2018).

Spatio-temporal paleoclimatic estimates are essential to drive process-based models that are capable of exploring causal mechanisms (Nogués-Bravo et al 2018). For instance, a recent ecological coupling study using climate emulation addressed the role of natural climate variability in shaping the evolution of species diversity in South America during the late Quaternary (Rangel et al 2018). That study used a paleoclimate emulator (Holden et al 2015) of the climate model PLASIM-ENTS (Holden et al 2014). The key limitations of the climate emulator were the lack of ocean dynamics in PLASIM-ENTS and the simplified emulation approach which only considered orbital forcing; large-scale approximations were made to account for the effects of time-varying ice sheets and $CO_2$. Here we address these weaknesses by using ensembles of a fully-coupled Atmosphere-Ocean GCM with varied orbit, ice-sheet and $CO_2$ boundary conditions. However, simulation alone would not be possible for an application of this ambition. We use the computationally-fast low-resolution AOGCM PLASIM-GENIE (Holden et al 2016), but even with this relatively simple model a five million-year transient simulation would demand ~300 CPU years of computing, which could not readily be parallelised. We overcome this intractability by using statistical emulation.

Emulators are computationally fast statistical representations of process-driven simulators, most useful when application of the simulator would be computationally intractable (Sacks et al 1989, Santner et al 2003, O'Hagan 2006). Climate applications of emulation have included the exploration of multi-dimensional parameter input space in order to, for instance, generate probabilistic outputs (Sanso et al 2008, Rougier et al 2009, Harris et al 2013) or calibrate simulator inputs (Sham Bhat et al 2012, Olson et al 2012, Holden et al 2013). Climate emulators have also been developed as fast surrogates of the simulator for use in coupling applications (Castruccio et al 2014, Holden et al 2014). In addition to Rangel et al (2018), coupling applications have included climate change impacts on energy demands (Labriet et al 2015, Warren et al 2018) and adaption to sea-level rise (Joshi et al 2016).

Our methodology uses principal component analysis to project spatial fields of model output onto a lower dimensional space of the dominant simulated patterns of change and then derives regression relationships between the simulator inputs and the coefficients of the dominant patterns. The method is analogous to the widely-used pattern-scaling technique (Tebaldi and Arblaster 2014), which assumes that an invariant pattern of simulated change can be scaled by global warming. Our approach extends this by including several (here ten) principal components for each climate variable, thereby allowing us to capture nonlinear patterns of change. The regression approach we use involves Gaussian process (GP) emulation (Rasmussen 2004).

GP emulators are non-parametric regression models that have become widely used tools in a variety of scientific domains. We train the emulators using boundary-condition ensembles of paleoclimate simulations, driven by variable orbital, $CO_2$ and ice-sheet forcing inputs, in order to predict spatial fields of bioclimatic variables as functions of these inputs. This builds on previous studies that have emulated two-dimensional climate fields from $CO_2$ forcing (Holden and Edwards 2010, Holden et al 2014), orbital forcing (Bounceur et al 2015, Holden et al 2015), from combined $CO_2$ and ice-sheet forcing (Tran et al 2016) and from combined orbital and $CO_2$ forcing (Lord et al 2017). Lord et al (2015) additionally considered two ice-sheets states (modern and a reduced Pliocene

configuration) but, to our knowledge, these three Pliocene-Pleistocene forcings have not previously been varied together except in the emulation of scalar indices (Araya-Melo et al 2015). Ice-sheet forcing complicates the emulation problem because ice sheets are three-dimensional input fields. Although climate emulators with
dimensionally reduced input and output fields have been developed (Holden et al 2015, Tran et al 2018), we simplify the problem by assuming there is an approximate equivalence between the ice sheet state and global sea-level. This reduces the emulation to the more usual problem of relating scalar inputs to high-dimensional outputs.

The motivation for our approach is to generate spatiotemporal climate fields for use in dynamic coupling
applications that need temporal variability and therefore cannot use snapshot AOGCM simulations. To this end, we need forcing time series that extend back 5 million years and have sufficient temporal resolution to capture orbitally forced climate variability. For PALEO-PGEM v1.0 we use the sea-level reconstructions of Stap et al (2017) for the whole period and their $CO_2$ reconstruction prior to 800,000 BP (when ice core records are not available).

**2 The model PLASIM-GENIE**
PALEO-PGEM was built from quasi-equilibrium simulations of the intermediate complexity AOGCM PLASIM-GENIE (Holden et al 2016), a coupling of the spectral atmosphere model Planet Simulator (PLASIM, Fraedrich 2012) to the Grid-Enabled Integrated Earth system model (GENIE, Lenton et al 2006). The component modules,
coupling and preindustrial climatology are described in detail in Holden et al (2016). PLASIM-GENIE is not flux corrected. The moisture flux correction required in the Holden et al (2016) tuning was removed during a subsequent calibration (Holden et al 2018). PLASIM-GENIE has been applied to studies on Eocene climate (Keery et al 2018) and climate-carbon cycle uncertainties under strong mitigation (Holden et al 2018).

We applied PLASIM-GENIE at a spectral T21 atmospheric resolution (5.625 degrees) with 10 vertical layers, and a matching ocean grid with 16 logarithmically spaced depth levels. We enabled the ocean BIOGEM (Ridgwell et al 2007) and terrestrial ENTS (Williamson et al 2006) carbon-cycle modules, as described in Holden et al (2018). We do not consider ocean biogeochemistry outputs here.

The 2000-year spun-up simulations required for emulation were performed with atmosphere-ocean gearing enabled (Holden et al 2018). In geared mode, PLASIM-GENIE alternates between conventional coupling (for 1 year) and a fixed-atmosphere mode (for 9 years), reducing spin-up time by an order of magnitude, to roughly four days CPU.

**3 Experimental overview**
We first provide a summary of the entire approach in five steps, as illustrated schematically in Figure 1. Each step is described in more detail in Section 4.

**i) Ensemble calibration:** We previously developed a 69-member ensemble of plausible parameter sets using
'history matching' (see, e.g., Williamson et al 2013). Applying any of these parameter sets to PLASIM-GENIE gives a reasonable climate-carbon cycle simulation of the present day, as evaluated by ten large scale metrics; all

69 parameter sets produce simulated outputs that lie within the ten history match acceptance ranges listed in Table 1. This step has been published elsewhere (Holden et al 2018).

**ii) Model selection:** We do not address parametric uncertainty in PALEO-PGEM, and so required a single favoured PLASIM-GENIE parameter set. One of the 69 history-matched parameter sets was identified by picking the parameter set whose simulator output had the largest likelihood (defined in Section 4.1) and this "optimised" parameter set was used in all subsequent simulations. We require PALEO-PGEM to describe glacial states and so, as part of the calibration, we performed an additional ensemble with the 69 parameter sets forced by Last

Glacial Maximum (LGM) boundary conditions. The calibration considered simulated LGM cooling in addition to the ten present-day metrics (Table 1).

**iii) Paleoemulator construction:** PALEO-PGEM was constructed via a two-stage process, in both stages applying Gaussian process emulation to a singular value decomposition of the outputs of a PLASIM-GENIE

simulation ensemble (c.f. Wilkinson 2010, Bounceur et al 2015, Holden et al 2015, Lord et al 2017). The first stage emulated the simulated climate response to variable orbital and $CO_2$ forcing, while the second stage emulated the incremental climate anomaly due to the presence of glacial ice sheets. The motivation for this two-stage approach was to impose physical meaning on the decomposition by isolating the ice-sheet forced components from the orbital and $CO_2$ forced components. Note that we do not assume a linear superposition of

the forcing components, and interactions between ice sheets, $CO_2$ and orbit are represented in the second stage (see Section 4.2). All simulations used the optimised parameter set, and varied only the climate forcing.

**iv) Paleoclimate emulation:** Forcing time series of orbital parameters, atmospheric $CO_2$ concentration and sea-level (as a proxy for ice-sheet volume) were applied to the two-stage emulator at 1000-year intervals to generate

emulated climates at the native climate model resolution.

**v) Downscaling.** The emulated climates were converted to anomalies with respect to the emulated preindustrial state and interpolated onto a high-resolution grid. These interpolated anomalies were applied to the observed climatology to derive a high-resolution paleoclimate reconstruction at 1000 year intervals from 5MaBP.


**4 The simulation ensembles**

**4.1 The optimised parameter set $\theta^*$**

Given computational constraints we chose to neglect parametric uncertainty in PALEO-PGEM, and selected a single 'optimised parameter set' for all simulations. Earlier work (Holden et al 2018) had developed a calibrated

ensemble of 69 plausible PLASIM-GENIE parameter sets through a history matching approach. In summary, these authors built and applied emulators of seven scalar metrics (items 1-7 in Table 1) to search for plausible input space. They considered hundreds of millions of potentially valid model parameterisations, each selected randomly by drawing from priors for 32 varied input parameters (Table 2). Each of these 32-element parameter vectors were applied to the seven emulators in turn and 200 of them were selected to maximize a criterion that

combined the distance of candidate points to the other points already in the design (to ensure the design points fully span the input space) and the probability (according to the emulator) of reasonably simulating the

observational targets: global average surface air temperature, global vegetation carbon, global soil carbon, Atlantic overturning circulation strength, Pacific Ocean overturning circulation strength, global average dissolved ocean oxygen concentration and global average calcium carbonate flux to the ocean-floor. The 200 parameter sets were applied to simulation ensembles of the preindustrial state and transient historical $CO_2$ emissions-forcing (1805 to 2005). Finally, 69 of these parameter sets were selected as acceptable on the basis of the seven pre-industrial metrics and three additional metrics that relate only to the transient simulations (items 8-10 in Table 1): emissions-forced $CO_2$ concentration in 1870 and 2005, and transient warming (from 1865 to 2005).

In addition to these ten plausibility tests of Holden et al (2018), we also required the optimized model to exhibit a reasonable response to glacial ice sheets. We therefore performed an additional 69-member PLASIM-GENIE ensemble, applying Last Glacial Maximum forcing of 180ppm $CO_2$ concentration, 'ICE-5G' LGM ice sheets (Peltier 2004) and the LGM orbital configuration of Berger (1989), with eccentricity 0.0019, obliquity 22.949° and longitude of the perihelion at vernal equinox 114.4°.

For each of j=1, …, 69 parameter combinations, we calculate a score $P_j$ which indicates how successful simulation j was, in terms of matching the observations for each of the eleven metrics. These are tabulated in the "Calibration" column of Table 1, where $\mu_i$ denotes the observational estimate for metric $i$ and $\sigma_i$ an estimate of uncertainty, cognizant of both observational and model error.

$$P_j = \prod_{i=1,11} e^{-(g_i(\theta_j)-\mu_i)^2/2\sigma_i^2} \tag{1}$$

where $g_i(\theta_j)$ is is the output of the simulator corresponding to the ith metric when it is run at parameter setting $\theta_j$. The optimised parameter set $\theta^*$ was selected to be the ensemble member with the highest score, equivalent to minimizing a weighted sum of squared errors. This optimised parameter set was used in all simulations that follow. The optimized output metrics are provided in Table 1, and the input parameter values in Table 2. The most notable bias is the cold LGM when compared to observational target, though the optimised model lies within the 3.1 to 5.9°C ranges simulated by the CMIP5/PMIP3 and PMIP2 ensembles (Masson Delmotte et al 2013).

The climate sensitivity of the optimised parameter set is 3.2°C. The maximum Atlantic overturning is 17.8Sv, at a depth of 1.1km with the 10Sv contour, an indicator of the location of NADW formation, at a latitude of 56°N. Under LGM forcing, Atlantic overturning weakens to a peak of 11.1Sv at a depth of 1.0km and the 10Sv contour shifts southward to 45°N. Under doubled $CO_2$ forcing, Atlantic overturning weakens substantially to a peak of 7.6Sv at a depth of 0.4km.

### 4.2 Ensemble design

Our approach to emulating climate output fields relies on dimension reduction using the singular value decomposition. This is a statistical technique which rotates the data onto a new orthogonal coordinate system, so that the first coordinate is in the direction of maximum variance in the data, the second coordinate is then in the direction of maximum variance conditional on being orthogonal to the first coordinate, etc. The new coordinates are often called principal components (or empirical orthogonal functions), and whilst they are orthogonal, they

are not expected to cleanly isolate distinct physical processes. In order to impose a physical separation of the components, and therefore to enforce a clean response to a distinct forcing, we chose to build the emulator as a two-stage process. We first decomposed and emulated the smoothly varying climate response to changing orbit and $CO_2$ concentration with fixed present-day ice sheets (the 'E1' emulator). The land-sea mask is held fixed at the present day in all simulations. We then separately emulated the incremental climate response to a change in ice-sheet state under the same orbital and $CO_2$ forcing (the 'E2'emulator) so that the final emulation is the sum of these two components.

To build the E1 and E2 emulators, two separate 50-member boundary-condition ensembles were performed (BC1 and BC2) with the optimized parameter set. The statistical design of both ensembles was the same 5x50 maximin latin hypercube (MLH,) varying the three orbital parameters, the $CO_2$ concentration and the ice-sheet state. The only difference between the two ensembles was that the fifth hypercube variable, reserved for ice sheets, was ignored for the BC1 ensemble and the present-day ice-sheet configuration imposed for all BC1 simulations. The BC1 ensemble is designed to simulate the model response to orbit and $CO_2$ forcing only, while the BC2 ensemble simulates the different response driven by the presence of glacial ice sheets under the same set of choices of orbital and $CO_2$ forcing.

The sampling strategy for the orbital variables (eccentricity $e$, the longitude of the perihelion at the vernal equinox $\omega$ and obliquity $\varepsilon$) followed Araya-Melo et al (2015), uniformly sampling $e \sin \omega$ and $e \cos \omega$ in the range -0.05 to 0.05 and $\varepsilon$ in the range 22° to 25°. This transformation was chosen because the insolation at any point in space and time of year is generally well approximated as a linear combination of these terms. Carbon dioxide was varied uniformly in log space, in the range log(160 ppm) to log(1000 ppm). For ice sheets, relevant only to the BC2 ensemble, four states were allowed in the training ensemble, being the Peltier Ice-5G ice sheets (Peltier 2004) at 10, 13 15 and 20ka. These times were chosen as they correspond to well-spaced ice-volume intervals as evidenced by benthic $\delta^{18}O$ (Lisiecki and Raymo 2007). These times correspond to sea-level falls of 29, 45, 64 and 107m relative to modern in the Stap et al (2017) reconstruction that we use to force the time series emulation (Section 6).

In contrast to Araya-Melo et al (2015), we did not restrict input space to exclude combinations of high $CO_2$ and high glaciation levels, preferring instead to use all BC1 ensemble members (i.e. including those with high $CO_2$) in the BC2 ice sheet anomaly ensemble. This maintained the maximin and orthogonal properties of the MLH design, and moreover avoided any risk of extrapolation outside of training input space during the Pliocene. Present day (~400ppm) $CO_2$ levels can be associated with significant (~50m) sea-level falls according to the Stap et al (2017) reconstructions (see Figure 2). However, the trade-off for this simplicity is that realistic input space during glacial periods was less well sampled than it would be for a more targeted ensemble of the same size (c.f. Araya-Melo et al 2015).

**5 Emulator construction**

Emulators were built for four bioclimatic variables: the mean temperature of the warmest and coolest quarters and the mean daily precipitation of the wettest and driest quarters. Each variable was calculated on a grid-point basis

as the maximum and minimum of the DJF, MAM, JJA and SON seasons. These emulated variables were chosen as being of bioclimatic relevance (c.f. Rangel et al 2018), and suitable for a wide range of ecological and impact coupling applications, defining the extremes of climate experienced over each grid-cell during a (decadally-averaged) annual cycle. Emulators of DJF and JJA temperature and precipitation were also built for validation purposes (Section 6.1).

We derived emulators from inputs of $e \sin \omega$, $e \cos \omega$, $\varepsilon$, $\log(CO_2)$ and sea level $S$, each normalized on the range -1 to 1. Sea level provides a proxy for ice-sheet volume, and hence ice-sheet state (under the assumption of an invariant correspondence between ice-sheets and sea level). This neglects the asymmetry of ice sheets under glaciation and deglaciation. The E1 emulator was built from the outputs of the BC1 ensemble (after centering the data, by subtracting the ensemble mean field $M$ from each simulation before singular value decomposition). The E2 emulator was built from the anomaly outputs BC2-BC1. For E2, we appended the training data with a synthetic 50-member ensemble with the hypercube inputs repeated except that sea level was randomly assigned to be between -25m and +100m. In these synthetic data, no simulations were performed, but instead all the climate anomalies were set to zero, equivalent to performing a second ice-sheet forced ensemble with a present-day ice sheets (and therefore with no anomaly by construction). This was needed so that the ice-sheet anomaly emulator can be used when glacial ice sheets are absent (i.e. sea level greater that -25m) i.e. when the ice-sheet emulated anomaly (E2) is trained to be zero and the emulation is determined only the orbit and $CO_2$ emulator (E1). Note that this approach neglected the loss of Antarctic and/or Greenland ice compared to modern that is implicit when paleo sea level exceeded the present day.

All emulators were built following the "one-step emulator" algorithm described by Holden *et al.* (2015), summarized briefly here. For each ensemble member, we formed the 2048-element vector which describes the 64 × 32 output field to be emulated. The vectors for the N ensemble members were combined into a (2048 × N) matrix **Y** describing the entire ensemble output of that variable. The matrices **Y** used to train the E1 emulators comprised decadal-averaged outputs of the BC1 ensemble, and these matrixes were centered by subtracting the ensemble mean field. The matrices for the E2 emulators were constructed from the decadal-averaged anomalies BC2-BC1. This separation of the forcing elements is a key difference with earlier work; every BC1 member has an identical BC2 member with the same inputs except for the incremental ice-sheet forcing, which cleanly isolates the emulation of ice-sheet forcing from the orbital and $CO_2$ forcing.

Singular value decomposition was performed to reduce the dimensionality of the simulation fields:

$$Y = LDR^T \tag{2}$$

where **L** is the (2048×N) matrix of left singular vectors ("components"), **D** is the N × N diagonal matrix of the square roots of the eigenvalues and **R** is the N × N matrix of right singular vectors ("component scores"). This decomposition produced a series of orthogonal components, ordered by the percentage of variance explained. We truncated the decompositions, considering only the first ten components. Each of the ten retained sets of scores thus comprised a vector of N coefficients, representing the projection of each simulation onto the respective

component. As each simulated field is a function of the input parameters, so are the coefficients that comprise the scores, so that each component score can be emulated as a scalar function of the input parameters to the simulator.

We used Gaussian process (GP) emulation (Rasmussen 2004) in preference to stepwise linear regression. The principal motivation for using this more sophisticated approach was that GPs are highly flexible non-parametric regression models which have greater modelling power than linear models. Linear models live in a finite dimensional space defined by polynomial functions of the covariates. Gaussian processes live in a much richer space of functions. An additional motivation was that GP emulation provides both a central estimate and an

estimate of uncertainty, and therefore provides us with a means to generate uncertain climate emulations in the absence of parametric uncertainty. It is important to note that emulator uncertainty is entirely distinct from (and therefore incremental to) parametric uncertainty.

**6 Emulator cross-validation and model selection**

Gaussian Process models are generalized models, but nevertheless require some user choices, the most important being the choice of covariance function. We used an anisotropic covariance function (different length scales for each input dimension) and estimated the unknown length scale parameters using the type II maximum likelihood estimators (Rasmussen and Williams, 2006). In order to evaluate the optimal covariance function, we considered the cross-validation metric $P$, see Section 4.3.1 of Holden et al (2014):


$$P = \sum_{c=1,10} R_c^2 V_c \tag{3}$$

where $R_c^2$ is the coefficient of determination of the emulator of principal component $c$, evaluated under leave-one-out cross-validation of all simulations, and $V_c$ is the variance explained by that component, summed across the

leading ten components. The metric is designed to quantify the percentage of the spatial variance explained by the emulator, capturing the explained variance due to principal component truncation (only ten components are considered) and to the emulation itself (i.e. the explained variance of the simulated component scores).

Table 3 summarises the cross-validation of the eight emulators (i.e. four bioclimatic variables, two forcing

categories). The second column tabulates the percentage of variance explained by the leading ten principal components, $\sum_{c=1,10} V_c$, and represents the maximum variance that could be explained by the emulators if they were perfect. The remaining columns tabulate the metric $P$ when building the emulator with a series of different covariance functions, being the alternatives available in the *DiceKriging* R package (Roustant et al 2012). The reduction in variance explained (relative to column 2) reflects additional errors due to emulation.


The temperature decompositions explain 94-99% of the ensemble variance, compared to 87-90% for the precipitation decompositions. Under emulation, the variance explained is 81-98% for the temperature fields and 73-83% for precipitation fields. The emulator performance is weaker for precipitation, because the low order components needed to explain much of the ensemble variability are more difficult to emulate.


The power exponential was found to give comparable or better performance compared to the other covariance functions in all eight emulators and was therefore chosen as the default covariance function, and used in all analysis that follows.

Table 4 summarises the variance explained under cross-validation of the seasonal and annual average emulators used in the following Section 7. DJF (JJA) temperature emulator performance is similar to Min (Max) temperature emulator performance, suggesting that northern hemisphere temperature is more difficult to emulate than southern hemisphere temperature, as would be expected for the ice-sheet emulator in particular. The performance of the various seasonal precipitation emulators is similar (82.7% to 84.8% for the orbit and $CO_2$ emulator, 72.4% to

75.4% for the ice-sheet emulator), but annual precipitation is easier to emulate than seasonal precipitation (88.6% for the orbit and $CO_2$ emulator, 81.9% for the ice-sheet emulator).

**7 Validation of reconstructed climate fields**

The emulators generate a paleoclimate as


$$E(e, \omega, \varepsilon, CO_2, S) = M + E1(e, \omega, \varepsilon, CO_2) + E2(e, \omega, \varepsilon, CO_2, S) \qquad (4)$$

where $M$ is the simulation mean field that was subtracted to center the ensemble before decomposition (Section 5). To generate a paleoclimate time series, we therefore require time series of the boundary condition

inputs $e, \omega, \varepsilon, CO_2$ and $S$.

For the orbital parameter inputs, we applied the 5 million-year calculation of Berger and Loutre (1991, 1999). We used $CO_2$ from Antarctic ice cores for the last 800,000 years (Luethi et al 2008). Prior to 800,000 BP, and for the entire sea-level record, we used the $CO_2$ and sea-level reconstructions of Stap et al (2017). These authors used a

zonally averaged energy balance model coupled to a 6-level ocean model, a thermodynamic sea-ice model and to one-dimensional mass-balance modules for each of the five major Cenozoic ice sheets (East and West Antarctica, Greenland, Laurentide and Eurasian). The Stap model is forced with benthic $\delta^{18}O$ records, and uses an inversing routine to de-convolve the temperature and ice-volume components of the isotope signal and generate a self-consistent time series of $CO_2$ and sea-level (ice volume).


Figure 2 plots the forcing time series and an illustrative application of the emulator, for which we emulated time-varying annual mean surface air temperature field and plot its area-weighted global average through time.

In order to validate the emulators, we performed a series of experiments with Mid-Holocene (MH), Last Glacial

Maximum (LGM), Last Interglacial (LIG) and Mid-Pliocene warm period (MPWP) $CO_2$, ice-sheets and orbital forcing. These time slices have been well-studied in Paleo-Modelling Inter-comparison Projects and are well suited to explore variability driven by all three forcings. The MH and LIG responses are dominantly forced by orbit, while the MPWP is dominantly forced by $CO_2$, and the LGM by both $CO_2$ and ice-sheet state.

**7.1 Mid-Holocene emulated ensemble**

To assist comparison with readily available PMIP2 data (Braconnot et al 2007), we here emulate seasonal (DFJ and JJA) fields rather than seasonal (MAX and MIN) fields, plotted in Figure 3. Uncertainty is associated with the emulation of the component scores. Gaussian process emulation quantifies this uncertainty by providing a mean prediction and an estimate of the uncertainty associated with that prediction. We generated a 200-member emulation ensemble with MH forcing. The 200 ensemble members differ because we do not assume the mean prediction for the emulated component scores, but instead draw randomly from the posterior distributions. In Figure 3 this ensemble is summarised with mean and standard deviation fields. (We note that for applications in which climate uncertainty is not addressed, it is appropriate to use the mean predictions of principal component scores to generate the best estimate.)

Fig 3 top panels compare emulated MH surface temperature (anomalies relative to preindustrial) with the PMIP2 OAV (coupled atmosphere-ocean-vegetation) ensemble. In northern winter DJF, high latitude warming is apparent in the emulated ensemble mean, although of uncertain sign (variability > mean). Cooling is apparent over all other land regions. In northern summer JJA, robust warming is apparent at mid to high latitudes, while changes of variable signs are apparent in the tropics, with cooling apparent over the Sahel, India and SE Asia. Each of these features is also found in the PMIP ensemble. The most significant difference is Antarctic cooling of ~3°C in PALEO-PGEM, which contrasts with a warming signal in the ensemble mean of PMIP2 (although we note DJF Antarctic cooling of 0.5°C was simulated in HadCM3M2). A significant cold Antarctic bias is also apparent during the Last Interglacial (Section 7.4). High southern latitudes are poorly modelled by PLASIM-GENIE. The preindustrial state exhibits a warm Antarctic bias, with greatly understated sea ice, a slow Antarctic Circumpolar Current and weak, northerly shifted zonal winds (Holden et al 2016), which are likely associated with well-known difficulties of resolving Southern Ocean wind stress at low meridional resolution (Tibaldi et al 1990, Schmittner et al 2010).

Fig 3 lower panels compare emulated MH precipitation with the PMIP2 OAV ensemble. In DJF, significant drying is emulated over central and northwestern South America, southern Africa, eastern Asia and northern Australia, while wetter conditions are emulated over northeastern South America. In JJA the largest changes are seen as a strengthening of the Asian monsoon precipitation, and significantly wetter conditions are also seen over the Sahel and western South America. These changes all reflect a general agreement with PMIP2.

**7.2 Last Glacial Maximum emulated ensemble**
We follow the emulated ensemble procedure for the Last Glacial Maximum. Fig 4 upper panels compare the emulated Last Glacial Maximum temperatures with the PMIP2 OA (ocean-atmosphere) ensemble. We neglect the OAV LGM ensemble because it has only two simulations. LGM cooling is dominated by cooling of up to ~40°C over the northern hemisphere glacial ice sheets. The most significant differences are apparent in the emulated uncertainty, which is understated by a factor of roughly two relative to PMIP. This is expected because the emulator is built from a single parameterization of PLASIM-GENIE and therefore does not capture uncertain climate sensitivity. We note that by applying the principles of invariant temperature pattern scaling (Tebaldi and

Arblaster, 2014), the temperature uncertainties due to neglected climate sensitivity could be approximated by
       inflating the variance of the principal component scores.

       Figure 4 lower panels compare emulated Last Glacial Maximum precipitation with the PMIP2 OA ensemble. In
       DJF, the drying apparent in central Africa, northern America and the Amazon are captured by the emulator, while
JJA drying at northern latitudes and in the Asian and African monsoon regions, and increased precipitation in
       South America are common to the emulator and the PMIP2 ensemble. The most significant difference is the
       increase of DJF precipitation emulated in central South America, which is not present in the PMIP ensemble
       mean, although we note that the PMIP2 simulations display change of uncertain sign.

**7.3 Glacial-interglacial variability**

       The emulated global temperature change over the last 800,000 years is plotted in Figure 5, reflecting the familiar
       glacial cycles and compared to the observationally based global temperature reconstructions of Koehler et al
       (2010). Ten separate emulators were built (following the steps described in Section 5 applied to annual average
temperature) and the mean prediction time-series for all ten emulators are plotted.

       The Last Glacial Maximum cooling across these ten emulators is $4.3 \pm 0.3°C$, which compares to $4.5 \pm 0.3°C$
       when emulated values were drawn randomly from a single emulator. The emulated estimates are lower than the
       simulated LGM cooling of 5.9°C (Table 1) and may reflect bias in the ice-sheet emulator under the extreme of
LGM forcing; the ice-sheet emulator was only able to explain 81% of the variance of cold season temperatures
       (Table 3). However, the seasonal patterns of emulated change are reasonable (Figure 4) and the annual average
       cooling is well-centered on the 3.1 to 5.9°C range simulated by the CMIP5/PMIP3 and PMIP2 ensembles (Masson
       Delmotte et al 2013).

Maximum warming of $0.3 \pm 0.1°C$ is emulated in the Last Interglacial (Marine Isotope Stage 5), peaking at 125
       kaBP. This is consistent with CMIP estimates of $0.0 \pm 0.5°C$, but lower than data-based estimates of ~1 to 2°C
       (Masson Delmotte et al 2013). Maximum warming in Marine Isotope Stage 11 is $0.1 \pm 0.2°C$, peaking at 401
       kaBP.

**7.4 Last Interglacial transients**

       Zonally-averaged emulated temperature changes are compared with the Last Interglacial transient model inter-
       comparison of Bakker et al (2013) in Figure 6 and Table 5. The latitudinal temporal trends are well captured by
       the emulator, considering the inter-model spread of Bakker et al (2013). Notably, temperatures in Jun-Jul-Aug
generally peak earlier (~125 kaBP) than temperatures in Dec-Jan-Feb (~120 kaBP). Maximum warming of ~2 to
       3°C is emulated in northern summer mid-high latitudes, peaking at 126kaBP, and consistent with inter-model
       estimates in the range 0.3 to 5.3°C, peaking between 125 and 128kaBP. Eight of the emulated peak warming
       estimates are consistent within the $1\sigma$ multi-model uncertainty ranges, and the remaining two are consistent within

$2\sigma$ multi-model uncertainty (Table 5). The clearest difference is seen in Antarctic summer, where cooling of up to 5°C is emulated, significantly greater than in any of the models.

### 7.4 The Mid-Pliocene warm period

The emulated climate of the Mid-Pliocene warm period is plotted in Figure 7. The only emulator forcing is $CO_2$ increased to 405ppm, as assumed in the model inter-comparison of Haywood et al (2013). Ice-sheets are fixed at present day, in contrast to Haywood et al (2013) where the boundary conditions included a reduced West Antarctic Ice Sheet.

Ensemble-averaged emulated warming is $1.6 \pm 0.2$°C and global precipitation change $0.10 \pm 0.01$ mm/day. These compare to multi-model estimates of 1.8 to 3.6°C and precipitation changes of 0.09 to 0.18 mm/day in Experiment 2 (the coupled atmosphere-ocean configuration) of Haywood et al (2013). Emulated high latitude warming of ~4°C is low-biased, but within the wide multi-model uncertainty range of ~3 to 14°C. Similarly, the emulated peak precipitation change of ~0.3 mm/day near the Equator is low biased, but within the multi-model range of ~0 to 1.3mm/day.

### 8 Downscaling

A spatial resolution higher than the native resolution of the underlying climate model may be required for paleo-applications given the scale dependency of many patterns and processes (e.g. Rahbek 2005), such as scale-dependent climate heterogeneity (Rangel et al 2018). We address this need by interpolating the low-resolution climate model anomalies onto fine-resolution climatological data. This approach is widely-used in climate impact assessment (e.g. Osborn et al 2016), and has also been applied in paleo-applications in anthropology (Melchionnaa et al 2018) and ecology (Rangel et al 2018).

Downscaling can be performed in any given grid. Here we illustrate downscaling on a global hexagonal grid build on a geodesic dome, because it minimizes geographic distortions in shape, area and distance that are common to map projections. The hexagonal grid is composed of 17,151 *quasi* equal-area cells of $6,918 \pm 859$ km$^2$ whose area variation is not spatially structured.

The four present-day (preindustrial) emulated bioclimatic variables $E0$ were linearly interpolated onto the geodesic grid. All emulations used the mean prediction and the E1 and E2 emulators were both truncated at ten principal components. Contemporary observations of the bioclimatic variables $C0$ were derived from WorldClim (Hijmans et al 2005), which provides temperature and precipitation estimates at 1 km$^2$ resolution, interpolated from temporally averaged measurements (1950 to 2000) from ~15,000-50,000 weather stations globally (depending upon the variable). The raw emulated climate data $E0$ and the difference with observed climatology $E0 - C0$ are illustrated in figure 8

The emulated climatology is reasonable, accepting the low resolution of the underlying climate model. Cold biases are generally confined to northern-winter high latitudes. Warm biases are more modest except for the Tibetan Plateau and Andes where the lapse rate cooling in these narrow mountain chains is poorly resolved by the climate

model (but corrected for by the downscaling approach described below). Excess precipitation bias is mostly
apparent in the (wet-season) monsoon regions. Deserts are generally well resolved in the emulator, a notable
exception being the hyper-arid Atacama, which is an orography-driven feature that cannot be captured at low
resolution. Conversely, orography-driven precipitation is understated in the Tibetan plateau. Precipitation is also
understated in the Sahel.

We apply anomaly adjustments to derive downscaled emulated climate fields through time $Ct$. This approach
preserves the high-resolution spatial heterogeneity of climatology. In the case of temperature this is
straightforward. Emulated anomalies $Et - E0$ are interpolated onto the hexagonal grid and applied additively, i.e.
$Ct = C0 + (Et - E0)$. For precipitation, the situation is more complex. In arid regions that are not well captured
by the emulator, a multiplicative anomaly approach is preferable $Ct = C0 \times (Et/E0)$, preserving hyper-arid
(topographically-forced) desert, and preventing unphysical negative precipitation when $Et - E0 < 0$.
Conversely, in wet regions that are understated by the emulator, a multiplicative anomaly approach can create
unphysically high precipitation, but an additive approach ensures a physically reasonable solution. A pragmatic
solution to this is to apply an additive precipitation anomaly when $E0 < C0$, and a multiplicative precipitation
anomaly when $E0 > C0$. This approach is well-behaved, noting that the additive and multiplicative anomalies are
equivalent when $E0 = C0$. Consider, when $E0 < C0$,

$$Ct = C0 + (Et - E0) > Et \tag{5}$$

and the additive anomaly partially compensates for the low bias in emulated climatological precipitation.
Conversely, when $E0 > C0$,

$$Ct = C0 \times (Et/E0) < Et \tag{6}$$

and the multiplicative anomaly partially compensates for the high bias in emulated climatological precipitation.

The present-day climatology and downscaled emulated LGM climate are illustrated in Figure 9. An animation of
the entire 5,000,000-year reconstruction is provided as supplementary material.

**9 Limitations of the approach**

PALEO-PGEM is to our knowledge the first attempt to provide a detailed spatiotemporal description of the
climate of the entire Pliocene-Pleistocene period. It is essential to understand the main limitations of our modelling
framework, discussed below, some of which may induce large errors or uncertainties in specific applications, or
even rule out certain applications completely. For all practical purposes and for the foreseeable future, substantial
uncertainties exist in any paleoclimate reconstruction as a result of incomplete knowledge, computing limitations
and irreducible climatic noise. Ideally, these uncertainties should be quantified in relation to any reconstruction
and their implications propagated through the analysis. Our approach provides an estimate of inherent uncertainty
derived from the emulation step of the reconstruction and thus underestimates the full uncertainty, but nevertheless

in some aspects remains comparable to the uncertainty in state-of-the-art reconstructions of particular periods as measured by the variance across ensembles of PMIP simulations.

Compared to state-of-the-art models, PLASIM-GENIE is a relatively low resolution, intermediate complexity climate model. This implies that processes operating at spatial and temporal scales below the native resolution of the climate model cannot be properly represented, although certain aspects of spatial variation are reintroduced in a highly idealised way by the downscaling process. The temporal effects of dynamical processes operating at sub-millenial timescales are further filtered out by the approximation inherent in the emulator construction that the climate is in quasi-equilibrium with the forcing, which is then only resolved at 1000-year time intervals.

In applications where (downscaled) time-slice simulations are adequate and are available from higher complexity models and/or multi-model ensembles (Section 7), these would normally be preferable to PALEO-PGEM as errors and biases will generally be smaller, particularly in high latitudes, regions of steep topography, close to coastlines or in known regions of locally extreme climate. We note that HadCM3 climate simulations (Singarayer et al 2017), downscaled to 1° resolution are available back to 120 kaBP (Saupe et al 2019), which would provide preferable (or supplementary) climate data for applications restricted to this time-domain.

The emulator uncertainty captures much of the uncertainty seen in multi-model intercomparisons (Figures 3 and 4), but PALEO-PGEM cannot fully represent model uncertainty, because it is derived from a single configuration of a single model. Most clearly in this respect, the 90% uncertainty range of climate sensitivity ($3.8 \pm 0.6$°C) is understated relative to multi-model estimates of $3.2 \pm 1.3$°C (Flato et al 2013). Some significant biases in spatial patterns are also apparent, most clearly temperature biases in high southern latitudes.

Emulator forcing is limited to orbit, $CO_2$ and ice sheets. Ice meltwater forcing is not considered so that millennial variability, especially important in North Atlantic, is neglected. The land-sea mask and orography are held fixed, so that ocean circulation changes driven by changing gateways (e.g. the closing Panama isthmus, with implications for the thermohaline circulation) are neglected and feedbacks driven by changing orography are neglected, especially important in regions of rapid tectonic uplift.

The representation of ice sheets applies Peltier 5G deglaciation ice sheets (Peltier 2004), assuming a fixed relationship between global sealevel reconstructions (derived from benthic oxygen isotopes) and the spatial form and extent of ice sheets. This approximation neglects the substantial asymmetry between build-up and decay phases of ice sheets and assumes that ice sheets were located similarly in all previous Pliocene-Pleistocene glaciations, which may not have been the case. Particular caution is therefore essential when applying the climate reconstruction at locations near to the margins of ice sheets.

We apply a downscaling approach because spatial climate gradients can be critically important for ecosystem dynamics, especially in mountainous regions which are poorly resolved at native climate model resolution (Rangel et al 2018). The downscaling approximation assumes that the lapse rate within a downscaled grid cell does not change with time, but it does capture the first order effect of topographic complexity by assuming a constant

present-day lapse rate. Similarly, the downscaling cannot capture feedbacks between atmospheric circulation and high resolution topography, which could alter the patterns of rain shadowing. However, for many applications, it is preferable to neglect this second order feedback than to neglect the first order effect of a rain shadow that could not be resolved at native climate model resolution (e.g. the Atacama), which downscaling imposes through the baseline climatology. Other simplifications include the implicit assumptions of fixed mountain glaciers and ecotone distributions. In short, the high-resolution reconstructions should not be interpreted as a faithful reconstruction of high-resolution climate, but serve to introduce a more realistic degree of spatial variability.

**10 Conclusions and summary**

We have used dimensionally reduced emulators of the intermediate complexity AOGCM PLASIM-GENIE, downscaled onto high resolution observed climatology, to generate a high resolution transient climate reconstruction of the last 5 million years. The reconstruction substantially improves on a previous emulated reconstruction (Rangel et al 2018) in the following ways

i) The underlying climate model is a fully coupled AOGCM. Rangel et al (2018) used PLASIM-ENTS (Holden et al 2014) which has a slab ocean and therefore neglected ocean circulation feedbacks.

ii) The new simulation ensembles considered climate forcing by orbit, $CO_2$ and ice-sheets. Rangel et al (2018) considered only orbit forcing, with large scale adjustments to crudely approximate the effects of $CO_2$ and ice sheets.

iii) We use Gaussian process emulation. Rangel et al (2018) used linear regression emulation, which cannot capture complex (non-linear) relationships between inputs and outputs.

These improvements allow us to provide a global emulation; the previous emulation was inappropriate for northern hemisphere due to the crude approximation of the response to ice sheet forcing. Additionally, we were able to extend the emulation back to 5 million years; the previous emulation was limited by the length of an existing 800,000-year transient GENIE simulations (Holden et al 2010) for $CO_2$ and ice sheets forcing. Finally, the use of GP emulation allows uncertainty estimates that we show in Figures 3 and 4 can be used to provide a reasonable proxy for model error, neglected in our single-parameterisation boundary condition ensembles.

The limitations of the reconstruction (see Section 9 for details) arise from the underlying climate model (low resolution, intermediate complexity), the approximated boundary conditions (in particular the use of only five ice-sheet states), uncertainties in the forcing time series (especially for sea level and $CO_2$), the assumption of quasi-equilibrium (so that e.g. millennial variability is neglected) and the limitations of downscaling. We note that the emulations and associated uncertainty compare favorably to existing ensembles of simulations with higher complexity models (Figures 3 and 4). We note further that reconstructing climates with different forcing time series is straightforward. Future improvements are anticipated by including a representation of changing topography. For instance, the Andes have uplifted by 25 to 40% of their 3,700m present day elevation over the last 5 million years (Gregory-Wodzicki 2000) and Himalayan uplift has been associated with intensification of the Asian monsoon about 3.6 to 2.6 Myr ago (Zhisheng et al 2001). Ensembles that address changing orography,

land sea masks and ocean gateways, will improve the simulated climate and allow the extension of the emulation further back in time, to periods in which it would be unreasonable to ignore tectonically driven change.

600

## 11 Code availability

The supplementary information contains the following

| | |
|---|---|
| PALEO-PGEMv1.0_5M_1Ka.mp4 | Animation of the four bioclimatic variables over 5Ma |
| PALEO-PGEMv1.0.R | R code to build and run the emulators. |
| **R input files** | |
| ensemble.dat | ensemble input design for the BC1/BC2 ensembles |
| 5000_1000_forcing.dat | time series forcing for 5Ma at 1kyr intervals |
| MH_forcing.dat | mid Holocene ensemble forcing |
| LGM_forcing.dat | Last Glacial Maximum forcing |
| area.dat | grid cell areas for area weighting |
| **data subdirectories** | |
| data | outputs of the BC1 PLASIM-GENIE ensemble |
| icedata | outputs of the BC2 PLASIM-GENIE ensemble |
| **supporting spreadsheets** | |
| ensemble | supporting calculations for the ensemble design |
| 5000ka_forcing | supporting calculations for the time series forcing |

620 PALEO-PGEMv1.0.R was saved with settings to emulate DJF temperature and produce a 5Ma time series using the GP mean prediction (no emulator uncertainty), ten principal components and a power exponential covariance function. Each of these settings can be changed as documented in the code. The code outputs the area-weighted average to screen, and three data sets to file: emul.dat (the full spatiotemporal output), mean.dat and SD.dat (the mean and standard deviation of the emulated fields, most relevant when code is set to generate an ensemble e.g.
625 with MH or LGM forcing).

## Author contributions

PBH, NRE and TFR developed the concept. PBH performed the PLASIM-GENIE simulations, using boundary conditions developed by GTT. PBH and RDW developed the GP emulators. TFR and EBP developed the
630 downscaling, with advice from PBH and NRE. PBH wrote the manuscript with contributions from all authors.

## Competing interests

The authors declare that they have no conflict of interest.

635

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

| $i$ | Output metric | Observations | History matching acceptance range | ML calibration (mean, 1sigma) $\mu_i, \pm\sigma_i$ | Optimized simulation $g_i(\theta^*)$ |
|---|---|---|---|---|---|
| 1 | Global average surface air temperature (°C) | ~14 Jones et al (1990) | 11 to 17 | 14 ± 1.5 | 14.1 |
| 2 | Global vegetation carbon (GtC) | 450 to 650 Bondeau et al (2007) | 300 to 800 | 550 ± 125 | 696 |
| 3 | Global soil carbon (GtC) | 850 to 2400 Bondeau et al (2007) | 750 to 2500 | 1625 ± 437.5 | 1170 |
| 4 | Maximum Atlantic Overturning (Sv) | ~19 Kanzow et al (2010) | 10 to 30 | 20 ± 5 | 17.8 |
| 5 | Maximum Pacific Overturning (Sv) | | <15 | 0 ± 7.5 | 2.4 |
| 6 | Global ocean averaged dissolved $O_2$ ($\mu$mol kg$^{-1}$) | ~170 Konkright et al (2002) | 130 to 210 | 170 ± 20 | 139 |
| 7 | Global deep ocean $CaCO_3$ flux (GT $CaCO_3$-C yr$^{-1}$) | ~0.4 Feely et at (2004) | 0.2 to 0.8 | 0.4 ± 0.15 | 0.56 |
| 8 | Atmospheric $CO_2$ in 1870 (ppm) | 288 Rubino et al (2013) | N/A | 288 ± 12.5 | 280 |
| 9 | Atmospheric $CO_2$ in 2005 (ppm) | 378 Keeling et al (2005) | 353 to 403 | 378 ± 12.5 | 380 |
| 10 | (1864-1875) to (1994-2005) warming (°C) | ~0.78 IPCC 2013 SPM | 0.6 to 1.0 | 0.78 ± 0.1 | 0.78 |
| 11 | Last Glacial Maximum temperature change (°C) | 4.0 ± 0.8 Annan and Hargreaves (2013) | N/A | -4.0 ± 1.2 | -5.9 |

**Table 1: Simulation output metrics for history matching and maximum likelihood calibration**


| Module | Parameter | Description | Units | Min | Max | Prior | Optimised $\theta^*$ |
|---|---|---|---|---|---|---|---|
| PLASIM | TDISSD | Horizontal diffusivity of divergence | days | 0.01 | 10 | LOG | 0.01245 |
| | TDISSZ | Horizontal diffusivity of vorticity | days | 0.01 | 10 | LOG | 0.04627 |
| | TDISST | Horizontal diffusivity of temperature | days | 0.01 | 10 | LOG | 1.03202 |
| | TDISSQ | Horizontal diffusivity of moisture | days | 0.01 | 10 | LOG | 0.06188 |
| | VDIFF | Vertical diffusivity | m | 10 | 1000 | LOG | 12.9576 |
| | TWSR1 | Short wave clouds (visible) | | 0.01 | 0.5 | LOG | 0.32403 |
| | TWSR2 | Short wave clouds (infrared) | | 0.01 | 0.5 | LOG | 0.03297 |
| | ACLLWR | Long wave clouds | $m^{-2}g^{-1}$ | 0.01 | 5 | LOG | 0.50152 |
| | TH2OC | Long wave water vapour | | 0.01 | 0.1 | LOG | 0.02357 |
| | RCRITMIN | Minimum relative critical humidity | | 0.7 | 1.0 | LIN | 0.94867 |
| | GAMMA | Evaporation of precipitation | | 0.001 | 0.05 | LOG | 0.00799 |
| | ALBSM | Equator-pole ocean albedo difference | | 0.2 | 0.6 | LIN | 0.44992 |
| | ALBIS[1] | Ice sheet albedo | | 0.8 | 0.9 | LIN | 0.8 |
| | APM[2] | Atlantic-Pacific moisture flux adjustment | Sv | 0.0 | 0.32 | LIN | 0.0 |
| GOLDSTEIN | OHD | Isopycnal diffusivity | $m^2s^{-1}$ | 500 | 5000 | LOG | 2005.24 |
| | OVD | Reference diapycnal diffusivity | $m^2s^{-1}$ | 2e-5 | 2e-4 | LOG | 1.35386e-4 |
| | ODC | Inverse ocean drag | days | 1 | 3 | LIN | 2.55463 |
| | SCF | Wind stress scaling | | 2 | 4 | LIN | 2.44654 |
| | OP1 | Power law for diapycnal diffusivity profile | | 0.5 | 1.5 | LIN | 1.07740 |
| BIOGEM | PMX | Maximum $PO_4$ uptake | mol $kg^{-1}$ $yr^{-1}$ | 5e-7 | 5e-5 | LOG | 2.27102e-5 |
| | PHS | $PO_4$ half-saturation concentration | mol $kg^{-1}$ | 5e-8 | 5e-6 | LOG | 1.21364e-6 |
| | PRP | Initial proportion POC export as recalcitrant fraction | | 0.01 | 0.1 | LIN | 0.031471 |
| | PRD | e-folding remineralisation depth of non-recalcitrant POC | m | 100 | 1000 | LIN | 802.258 |
| | PRC | Initial proportion $CaCO_3$ export as recalcitrant fraction | | 0.1 | 1.0 | LIN | 0.22708 |
| | CRD | e-folding remineralisation depth of non-recalcitrant $CaCO_3$ | m | 300 | 3000 | LIN | 1315.25 |
| | RRS | Rain ratio scalar | | 0.01 | 0.1 | LIN | 0.076452 |
| | TCP | Thermodynamic calcification rate power | | 0.2 | 2.0 | LIN | 0.510763 |
| | ASG | Air-sea gas exchange parameter | | 0.3 | 0.5 | LIN | 0.46006 |
| ENTS | VFC | Fractional vegetation dependence on carbon density | $m^2$ $kgC^{-1}$ | 0.1 | 1.0 | LIN | 0.84249 |
| | VBP | Base rate of photosynthesis | kgC $m^{-2}$ $s^{-1}$ | 9.5e-8 | 2.2e-7 | LIN | 1.2040e-7 |
| | LLR | Leaf litter rate | $s^{-1}$ | 2.4e-9 | 8.2e-9 | LIN | 2.4197e-9 |
| | SRT | Soil respiration temperature dependence | K | 197 | 241 | LIN | 218.356 |
| | VPC[3] | $CO_2$ fertilization Michaelis-Menton half-saturation | ppm | 29 | 725 | LOG | 215.368 |

**Table 2: Prior distributions for PLASIM-GENIE varied parameters (uniform between ranges in log/linear space as stated). Notes. 1) ALBIS ice sheet albedo was fixed at 0.8 in the final ensemble. 2) APM was fixed at zero in the final ensemble (no flux correction). 3) VPC was not constrained by the emulator filtering as this parameter has no effect in the preindustrial spin up state. The final calibration step, selecting 69 simulations that satisfy present-day plausibility after the historical transient was primarily an exercise to calibrate the VPC parameter. Prior distributions are discussed and derived from Holden et al (2010, 2013a, 2013b, 2014 and 2016). The final column tabulates the optimised parameter set.**



| | PC variance explained | Matern 3/2 | Matern 5/2 | Gaussian | Exponential | Power exponential |
|---|---|---|---|---|---|---|
| **Orbit and CO$_2$ emulator** | | | | | | |
| Max precipitation | **89.7%** | 81.7% | 80.2% | 76.9% | 81.0% | **82.7%** |
| Min precipitation | **87.2%** | 81.9% | 80.9% | 78.3% | 81.8% | **82.7%** |
| Max SAT | **99.3%** | 97.7% | 97.3% | 96.8% | 97.8% | **98.1%** |
| Min SAT | **99.5%** | 95.1% | 95.2% | 95.3% | 95.2% | **95.0%** |
| **Ice-sheet emulator** | | | | | | |
| Max precipitation | **88.5%** | 74.7% | 72.5% | 67.0% | 71.9% | **75.4%** |
| Min precipitation | **88.4%** | 72.1% | 69.3% | 60.7% | 69.4% | **73.3%** |
| Max SAT | **98.7%** | 94.2% | 93.6% | 92.3% | 95.1% | **95.1%** |
| Min SAT | **98.0%** | 79.3% | 77.5% | 74.6% | 80.8% | **80.9%** |

**Table 3. Optimization of the Gaussian process covariance function. The variance explained by the first ten components of the decomposition is quantified by "PC variance explained", which would be the expected variance explained if the emulators were perfect. The percentage of variance explained by the emulators is quantified by the metric P (Eq. 3, including ten components) for each of the eight emulators, considering various tested covariance functions. A power exponential is favored for the final emulator, having similar average performance to exponential covariance function, but outperforming it for the more difficult precipitation variables.**

| | DJF | JJA | Max | Min | Mean |
|---|---|---|---|---|---|
| Orbit and CO$_2$ emulator | | | | | |
| Precipitation | 84.8% | 83.9% | 82.7% | 82.7% | 88.6% |
| SAT | 95.0% | 97.8% | 98.1% | 95.0% | 96.7% |
| Ice-sheet emulator | | | | | |
| Precipitation | 74.0% | 72.4% | 75.4% | 73.3% | 81.9% |
| SAT | 82.1% | 94.8% | 95.1% | 80.9% | 90.4% |

**Table 4. Seasonal and annual mean emulator performance (as used in Section 7), measured by the metric P (Eq. 3, including ten components). A power exponential covariance is used in all cases. Note that max and min values repeat data from Table 3.**

| | 60°N-90°N | 30°N-60°N | 30°S-30°N | 60°S-30°S | 90°S-60°S |
|---|---|---|---|---|---|
| DJF peak warming °C | 1.7 (-5.8 to 1.2) | 0.1 (-0.8 to 2.1) | 0.4 (0.6 to 1.2) | 0.4 (-0.7 to 1.0) | -0.3 (-1.3 to 2.3) |
| DJF year of peak warming BP | 124 (118 to 124) | 119 (117 to 121) | 119 (116 to 119) | 119 (119 to 121) | 119 (116 to 118) |
| JJA peak warming °C | 2.4 (0.3 to 3.7) | 3.0 (0.7 to 5.3) | 0.7 (0.3 to 2.5) | 0.2 (-0.7 to 1.0) | -0.3 (-1.3 to 2.3) |
| JJA year of peak warming BP | 126 (125 to 128) | 126 (126 to 129) | 125 (127 to 130) | 118 (124 to 130) | 119 (126 to 129) |

**Table 5. Last Interglacial peak warming (°C) and year of peak warming (BP) compared to the model inter-comparison $\pm 1\sigma$ ranges of Bakker et al (2013). Emulated data are provided for Dec-Fan-Feb and Jun-Jul-August, compared to January and July data in the model inter-comparison, and comparisons are provided for five latitude bands.**

**Figures**

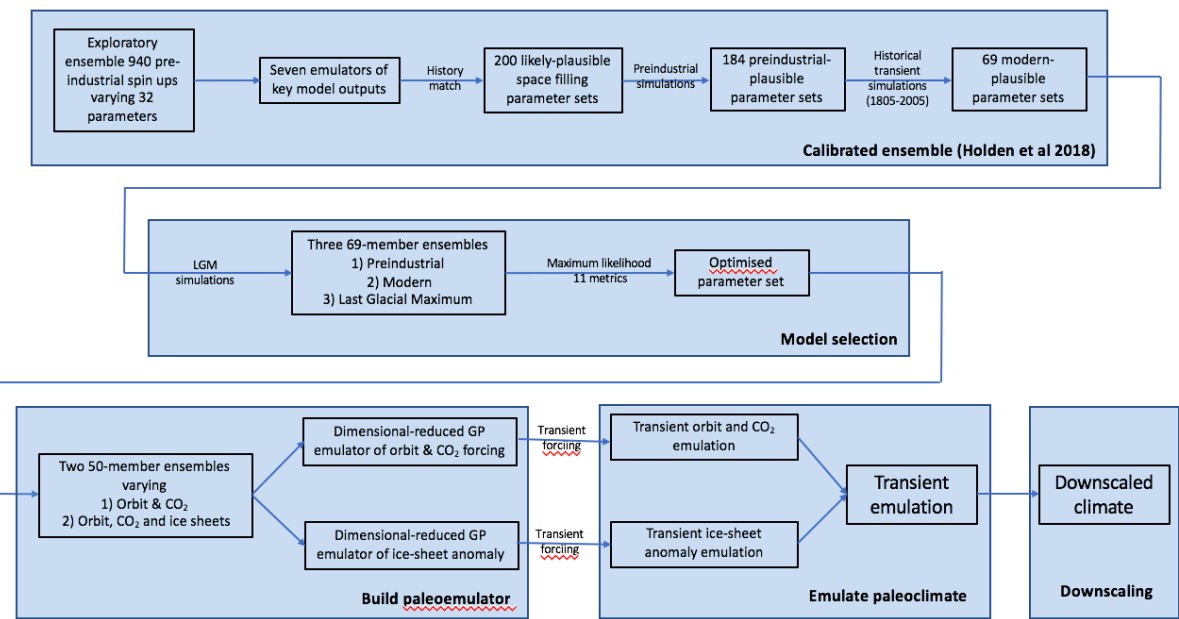

**Figure 1: Schematic of experimental design**


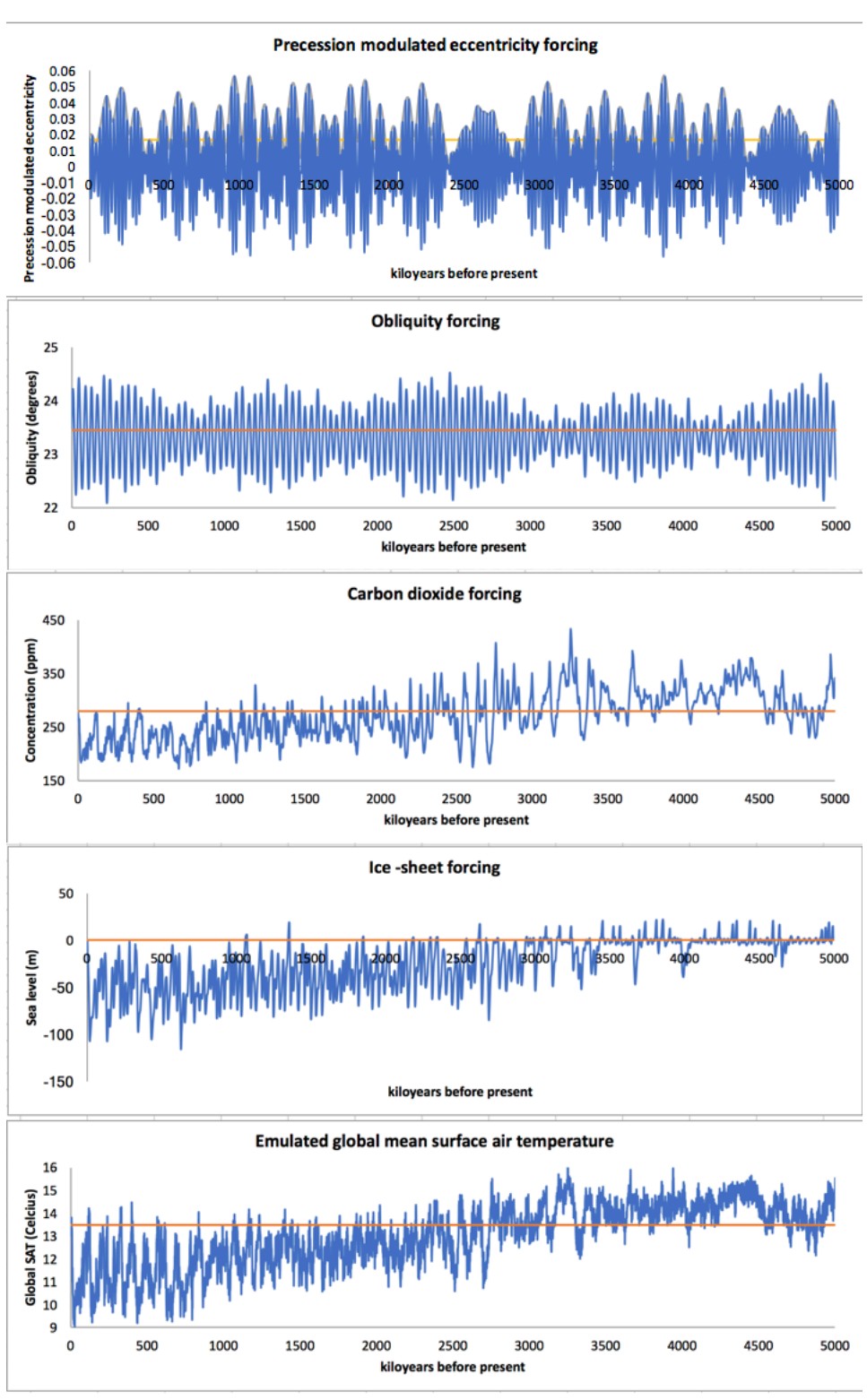

**Figure 2: Emulator time-series forcing and reconstructed global surface air temperature. Orbital forcing is Berger and Loutre (1991, 1999). Ice-sheet forcing is the sea-level reconstruction of Stap et al (2017). Carbon dioxide forcing after 800,000 years BP is ice-core data (Luethi et al 2008), using the Stap et al (2017) reconstruction in the earlier period.**


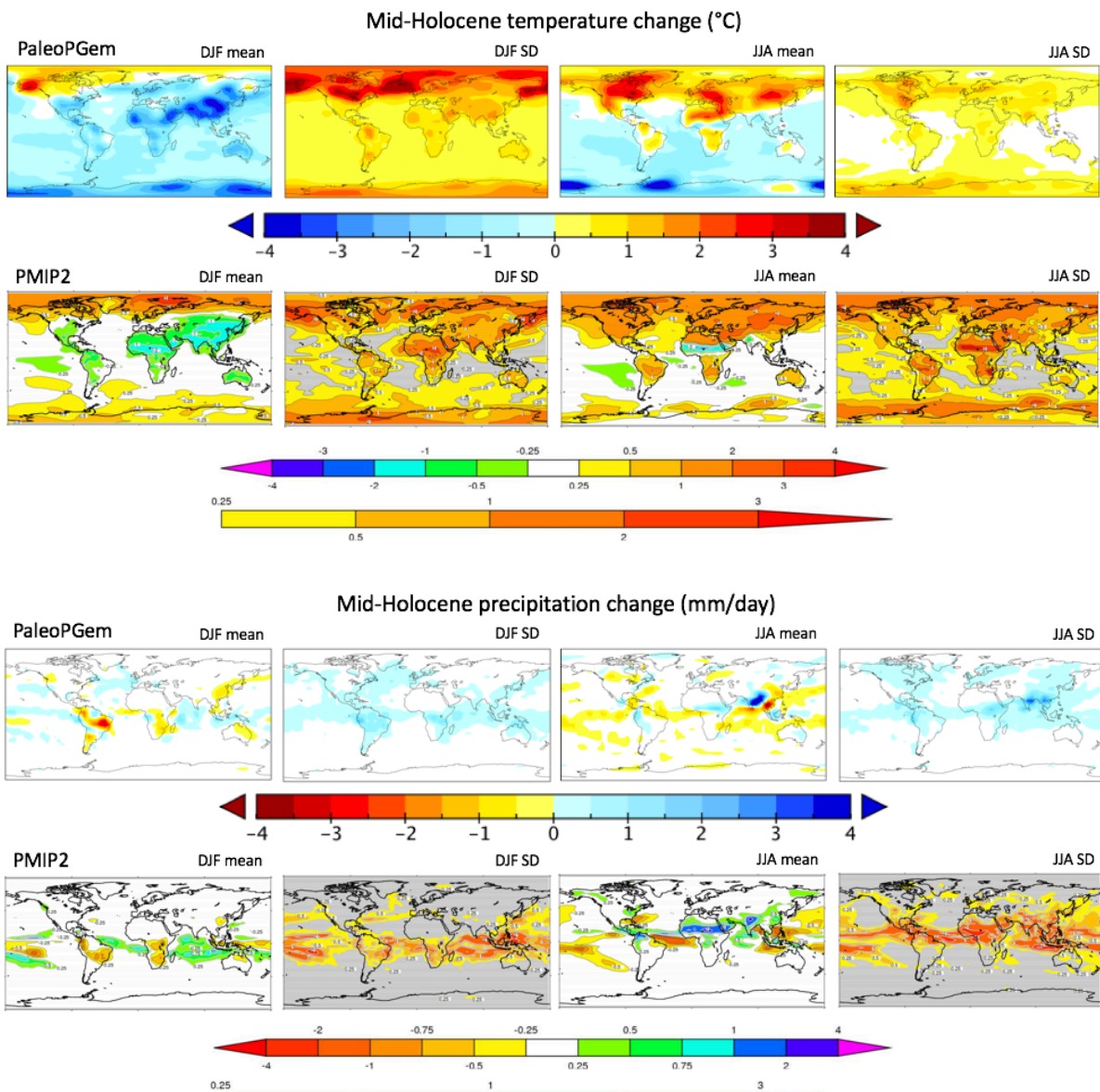

**Figure 3: PALEO-PGEM emulated ensemble comparison with PMIP2 Ocean-Atmosphere-Vegetation Ensemble (Braconnot et al 2007) for the mid Holocene.**



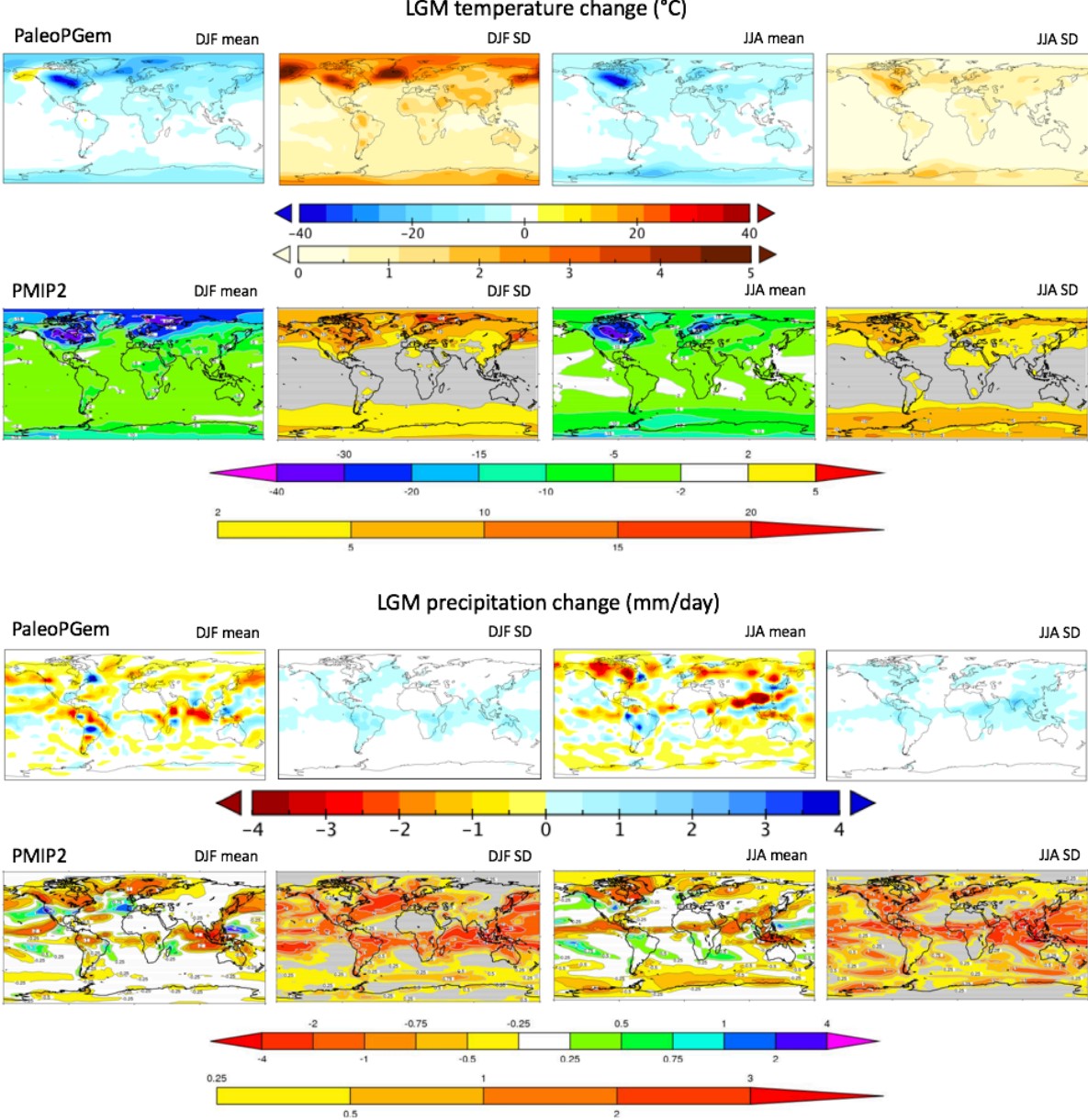

**Figure 4: PALEO-PGEM emulated ensemble comparison with PMIP2 Ocean-Atmosphere Ensemble (Braconnot et al 2007) for the Last Glacial Maximum. Note the different scales for SD temperature. Reduced variance in PALEO-PGEM is due to the understated uncertainty of climate sensitivity, which arises from the neglect of parametric uncertainty.**


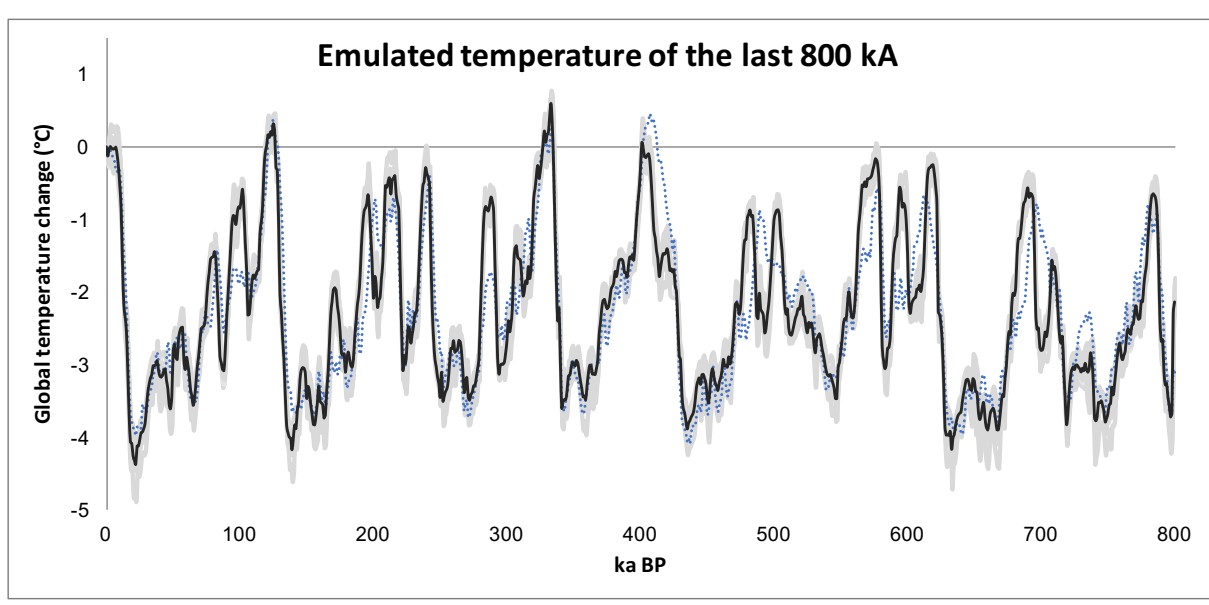

**Figure 5: Emulated global temperature over the last 800,000 years. An emulator was built ten times and the mean prediction time series of each emulator are plotted as grey lines, with the mean of these plotted as the single black line. The blue dotted line is the observationally based reconstruction of Koehler et al (2010). "Inter-emulator" variability compares to emulated LGM ensemble cooling (i.e. when drawing principal components scores randomly from a single emulator) of 4. 5 ± 0. 3 °C.**


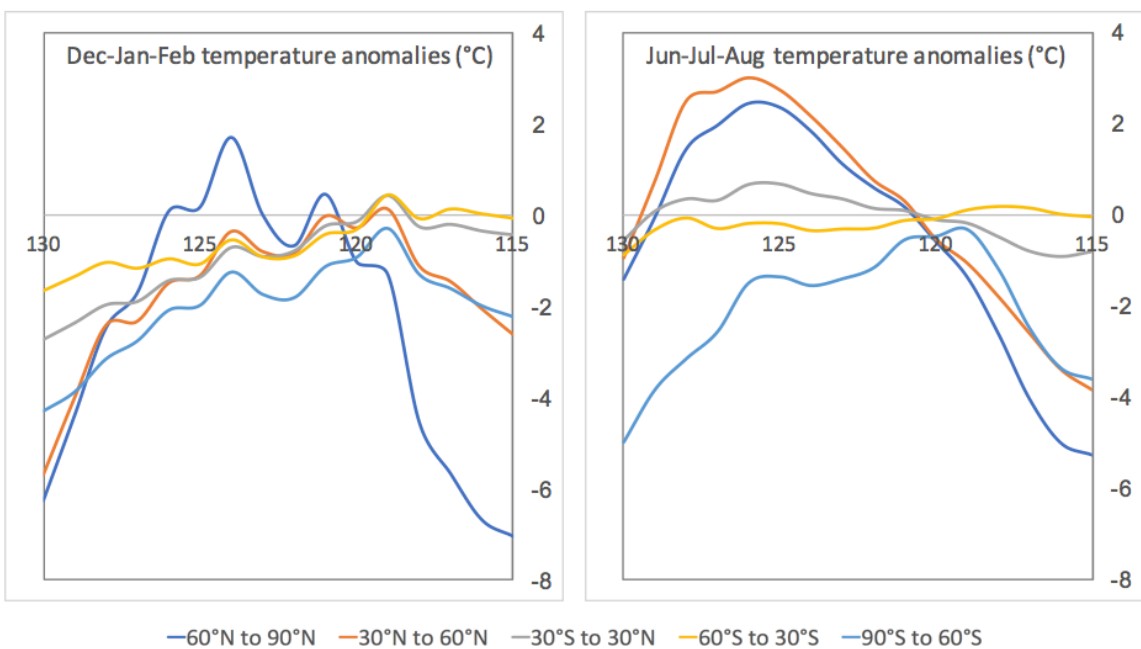

**Figure 6: Emulated Last Interglacial temperature anomalies with respect to pre-industrial. Data are provided for Dec-Jan-Feb and Jun-Jul-Aug averaged over five latitude bands c.f. Figures 2 and 3 of Bakker et al (2013).**


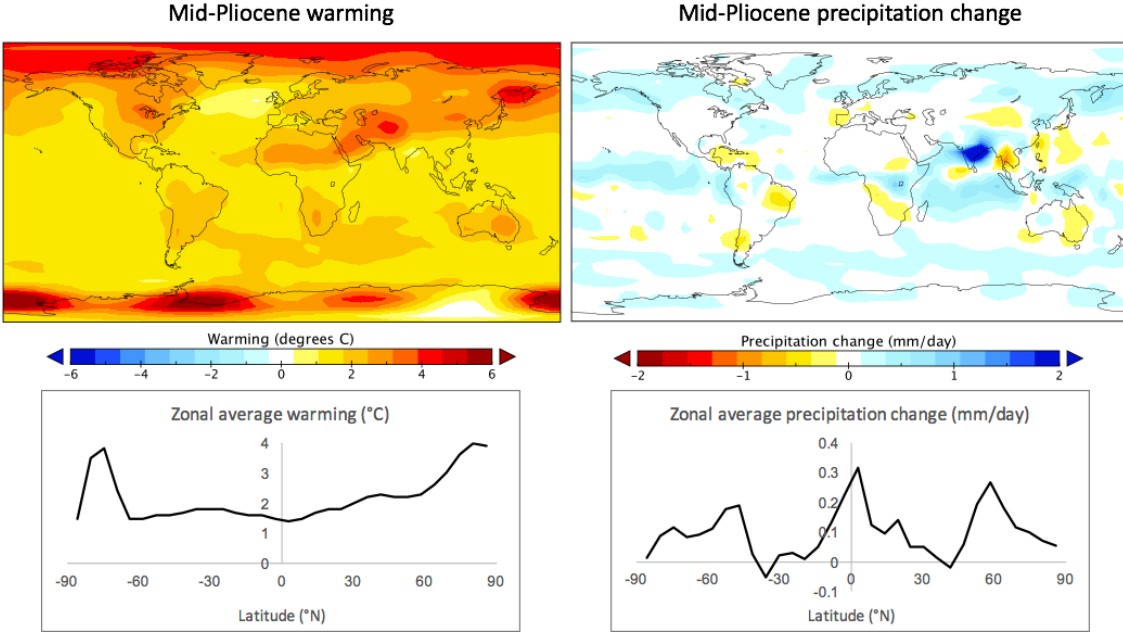

**Figure 7: Emulated Mid-Pliocene temperature and precipitation anomalies with respect to pre-industrial. The ice-sheet and orbital inputs are set to preindustrial, and the emulated change is driven by an assumed CO₂ concentration of 405ppm**

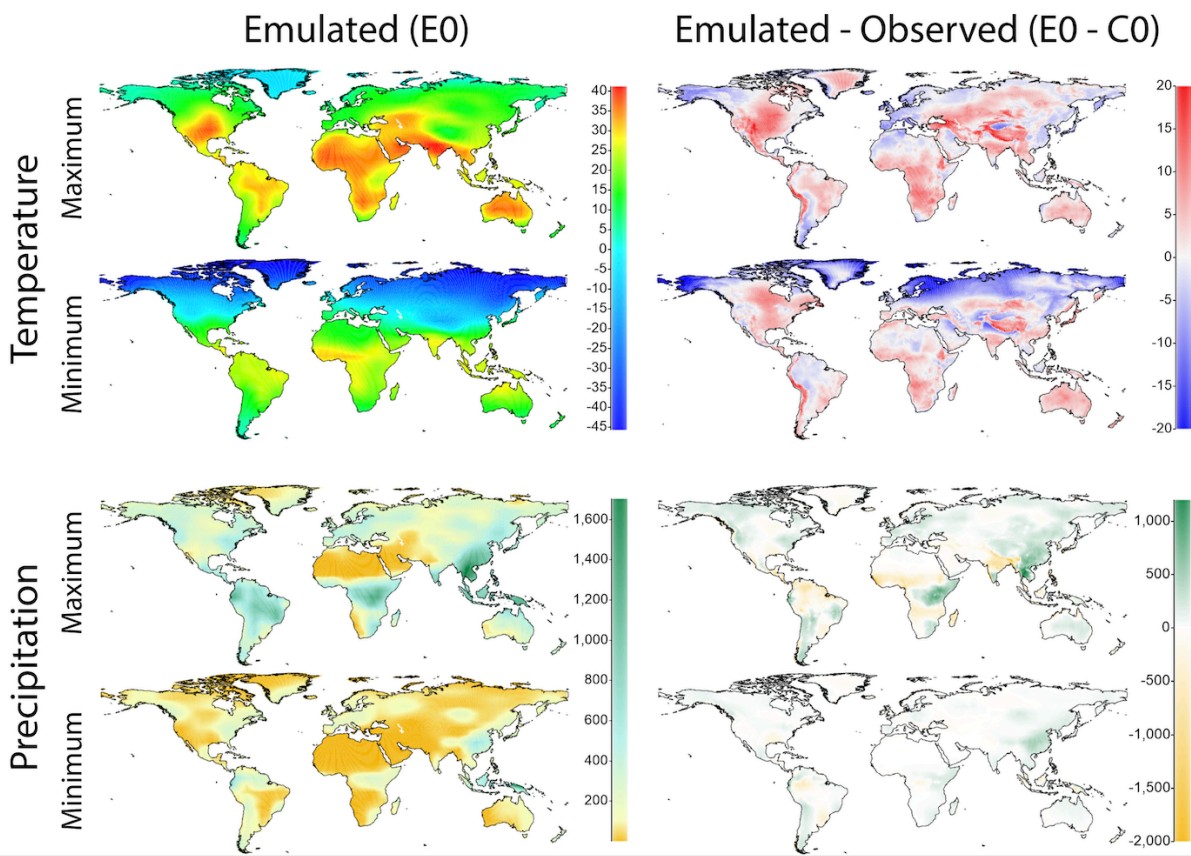

**Figure 7: Downscaling the emulated climate. Left hand panels are the preindustrial emulations of the seasonal bioclimatic variables at native (T21) model resolution, interpolated onto the high-resolution grid. Right hand panels illustrate the differences with respect to high resolution climatology (Hijmans et al 2005).**


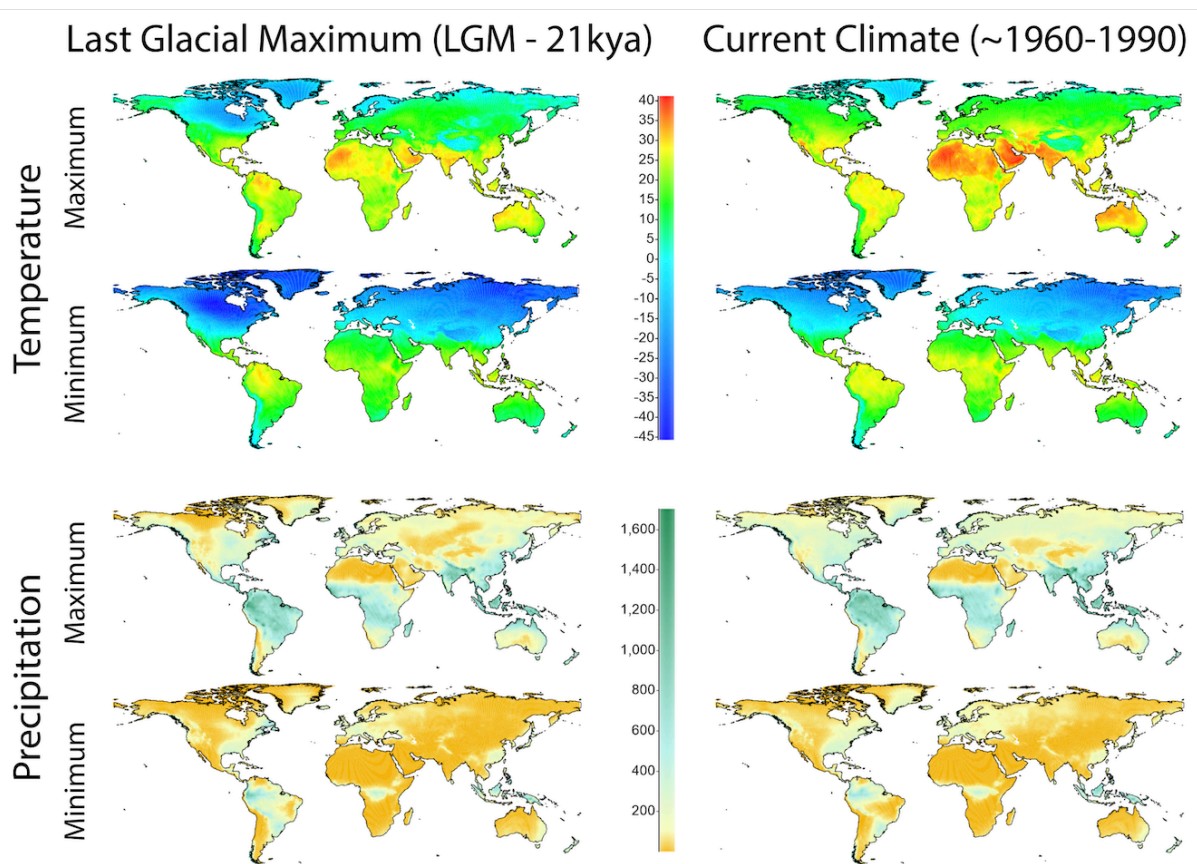

**Figure 8: Downscaled emulated climate. Left hand panels are the downscaled emulated bioclimatic variables at the Last Glacial Maximum. Right hand panels are the present-day climatology (Hijmans et al 2005). Note that downscaled climates are derived by applying emulated anomalies to this present-day climatology. An animation of the complete 5 Ma reconstruction is provided as supplementary material.**