# Peer review of "PALEO-PGEM v1.0: A statistical emulator of Pliocene-Pleistocene climate"

_Geoscientific Model Development, 2018_

## Referee Comment (RC1) · Crucifix (Referee) · 18 Jan 2019

The authors propose a latitude-longitude reconstruction of the climate of the whole Pleistocene, using a Gaussian process emulator calibrated on two experiment designs with the PLASIM-GENIE model. It uses use $CO_2$ and sea-level as inputs, based on an inverse modelling reconstruction provided by Stap et al. and, where available, ice core observations. R code with input files are provided.

The process for designing and calibrating the emulator is largely based on earlier work (experiment design, PCA emulator). There is however a cunning novelty: using two similar experiment designs for isolating the climate anomaly caused by ice sheets.

Although perhaps not in line with the reviewer 'etiquette', I wish to make a personal

comment, in the hope that the editor and authors will be forgiving find this intrusion useful for the evaluation of the work under concern here.

After the articles of Araya-Melo et al. (2015), Bounceur et al. (2015), and Lord et al. (2017), we had everything in place at UCLouvain to provide a similar reconstruction, and in fact we tried a few. What stopped us from publishing are:

- that the ocean circulation in LOVECLIM was not behaving adequately, with stronger, deep ocean circulation at glacial maxima, at odds with proxies for ocean ventilation. At some stage we thought of mending the simulation with an additional freshwater perturbation, but this work was never finalised to the point of publishing.

- the strategy used in Araya-Melo et al. 2015 of summarising ice sheet forcing with a single quantity, and which is applied here by Holden et al., can be problematic. There is nothing to guarantee that Weschelian ice sheets were located similarly to those of the Early Pliocene, and also, as the authors rightly acknowledged, the build-up and decay phases of ice sheets are quite asymmetrical during the late Pleistocene. This might not be that much of a problem in certain applications, but it can be heavily misleading to users who would use this product in Europe or in Siberia without much discernment. For example, I would be particularly worried of archaeologists using the provided reconstructions near the limits of ice margins. A 3-D reconstruction of the Pleistocene can be very popular, so it needs to be disseminated wisely.

This little experience brings me to the following, and related comments, about the present contribution by Holden et al.

1. It definitely needs to come up with appropriate health warnings about usage limits of the reconstruction. This is particularly crucial since the introduction presents it

a product oriented to end-users including archaeologists and biodiversity experts. A critical evaluation of the validity of the reconstructions near the North Atlantic (with emphasis on ocean circulation effects) needs to be provided.

2. In the introduction it is clearly said that uncertainty attached to the emulator (here, as a surrogate of GENIE-PLASIM) is distinct from the model uncertainty. This is true, and hence what would think that the evaluation (or "validation") of the emulator (as a surrogate) should be clearly distinguished from the evaluation of the model as a representation of the real climate. I found that this important distinction is pretty blurred in the section 6 (strangely divided into a section heading an a subsection 6.1). In fact there is very little about the evaluation of the emulator as a surrogate of GENIE-PLASIM. The authors refer to the PMIP ensemble and feel comfortable that the emulator-based reconstruction is in broad agreement with PMIP simulation of the mid-Holocene and the LGM, but in doing this the authors are mainly evaluating the reconstruction, not the emulator as a statistical surrogate. And, as I suggest in point 1. above, this evaluation is not providing with non-climatologist users with enough information about its application domain (the dos and don'ts). It is also quite uncomfortable that the emulator provides so-called bioclimatic variables (MIN and MAX over the seasonal cycle) while the "validation" is made on the basis of seasonal averages.

3. Downscaling. Is this correct that downscaling as presented here assumes constant sub-grid correction anomaly, defined as the difference between the present-day observations and simulated grid-box-mean in a reference experiment (the anomaly being on the log of precipitations in the low-precipitation areas)? This treatment is arguably inadequate in palaeoclimate applications, where topography, surface type (think of Swiss glaciers to take but one example), ice sheet margins, land-sea mask, and ecotone boundaries vary substantially. Again, aren't we misleading the users by providing the illusion of a high-res reconstruction, while it may in fact be quite wrong at places? For reference, Levavasseur and co-authors

have provided some thoughtful contributions about downscaling in palaeoclimate applications (e.g.: Levavasseur et al., 2010, The Cryosphere, 10.5194/tcd-4-2233-2010, and subsequent references).

**0.1 Minor comments**

- L. 280-295: Define the R-squared score and clarify what is mean by "the performance averaged over the eight emulators" (since the P-metric is near one, an arithmetic mean of P may be inadequate).

- Clarify the approach implemented for calibrating the length-scales appearing in covariance functions: do they vary across components, across variables?

- The original reference for Berger and Loutre, as quoted here, is Berger and Loutre (1991), doi 10.1016/0277-3791(91)90033-Q . It could be cited along with the Pangea reference.

- Figures 3 and 4 are a bit overwhelming, with small character size, and only the one with eyes trained in deciphering PMIP-type experiments will understand that the anomalies seen here are reasonably expected from GENIE-PLASIM and understand its limits.

**0.2 Conclusion**

The article could be a nice addition to current efforts in simulating the Pleistocene climate, but it is ambiguous as to its objective. If the authors ambition is to provide an technical, significant improvement on emulation, then they need to focus more on the evaluation of the emulator as such, and be more thorough in the discussion of the different technical options. If the ambition is to provide a final product to be used by

non-climate users, then I would urge the authors to be much more critical about the pitfalls of the current reconstruction, and in the present state, I would actually discourage dissemination of this product, since the risks of it being misused are too large.

---

## Referee Comment (RC2) · Anonymous Referee #2 · 23 Apr 2019

First, I apologize to the authors for taking so long to complete this review of "PALEO-PGEM v1.0: A statistical emulator of Pliocene-Pleistocene climate" for publication in GMD. In this paper, the authors describe the building of a statistical emulator from the PLISM-GENIE intermediate complexity model in order to provide a continuous reconstruction of important climate variables over the last 5 millions years, using interpolation to downscale these variables to a higher resolution. This paper logically follows a number of previous articles describing both building of emulators of intermediate complexity models (e.g., Holden et al. 2014) and their application to a range of important scientific questions, such as the links between climate and biodiversity (e.g. Range et al. 2018). It adds in particular a representation of ocean dynamics, which was lacking in previous studies, via the use of PLASIM-GENIE, the coupling of the two models having been

described previously (Holden et al. 2018). Overall, the paper is well written and the methodology is sound. I thus recommend publication provided that some important points are taken care of or better discussed:

1. There is no evaluation of the impact of the addition of ocean dynamics on the climate reconstructions, despite being one of the main justifications of the paper (l. 48-49, l. 430). How different would be the results using PLASIM-ENTS rather than PLASIM-GENIE? This leads me to another question: in the PLASIM-GENIE experiments with varying ice sheets, is the land/sea mask changed accordingly or are only the height/albedo feedbacks taken into account (with no varying coastlines impacting ocean dynamics)? If the land/sea mask is modified, I wonder how valid are the transition between ocean modes linked to varying sea level in the subsequent emulations of the climate?

2. Related to the above, the anomaly method for the downscaling approach supposes that the biases between an observed climate state Ct and the emulated climate Et remain somehow constant through time. This is a very crude assumption given, among others, the documented changes in ocean circulation dynamics across the last 5 Ma. How well is the ocean circulation represented in the model? And ocean circulation changes, e.g. between the LGM and MH?

3. I am a bit disappointed by the lack of actual comparison to data over the last 5 Ma. For instance, how does it represent the mid-Pliocene period (and compare to the extensive database of PlioMiP exercises) or glacial-interglacial T and P variations? And transient events like MIS M2?

4. I think a concise but very general reminder of what is an emulator and of the theoretical basis behind could be useful for non specialized readers. In the present version, this reduces to one sentence (l. 58-59) because the Sections 3 and 4 either use too many reference to previous papers (Section 3) or are probably too specialized already (Section 4).

5. The discussion on the limits of the approach and of the (numerous) uncertainties and approximations made should be expanded, in particular if the aim is to provide a widely available climatic reconstruction that is then used to force ecological niche modelling or biodiversity models (l. 36-42). Because this requires at least some confidence in the ability of the model to reproduce "true" (absolute) paleoclimatic conditions and variability (see points 1-3 above).

Other minor details l. 52 "naïve simulation would not be possible for an application of this ambition". I do not understand this sentence.

l. 272-274. "GPs are highly flexible non-parametric regression models which have greater modelling power than linear models". Please clarify.

l. 393. "Warm biases are more modest". On the basis of the figure, this is hard to believe as there are regions with a warm bias as large as the cold bias of other regions.

Fig. 3 and 4. Please use the same color scale and range as the PMIP ensemble to ease comparisons. Why is the Southern Ocean cooling rather than warming in the MH simulation?

---

## Author Comment (AC1) · 18 Jun 2019

Crucifix (Referee)  michel.crucifix@uclouvain.be

**We thank Michel Crucifix for this thorough and useful review. Our responses are in bold face and the associated revisions to the manuscript are in italics.**

The authors propose a latitude-longitude reconstruction of the climate of the whole Pleistocene, using a Gaussian process emulator calibrated on two experiment designs with the PLASIM-GENIE model. It uses use $CO_2$ and sea-level as inputs, based on an inverse modelling reconstruction provided by Stap et al. and, where available, ice core observations. R code with input files are provided.

The process for designing and calibrating the emulator is largely based on earlier work (experiment design, PCA emulator). There is however a cunning novelty: using two similar experiment designs for isolating the climate anomaly caused by ice sheets.

Although perhaps not in line with the reviewer 'etiquette', I wish to make a personal comment, in the hope that the editor and authors will be forgiving find this intrusion useful for the evaluation of the work under concern here.  After the articles of Araya-Melo et al. (2015), Bounceur et al. (2015), and Lord et al. (2017), we had everything in place at UCLouvain to provide a similar reconstruction, and in fact we tried a few. What stopped us from publishing are:

- that the ocean circulation in LOVECLIM was not behaving adequately, with stronger, deep ocean circulation at glacial maxima, at odds with proxies for ocean ventilation. At some stage we thought of mending the simulation with an additional freshwater perturbation, but this work was never finalised to the point of publishing.

**We have added a paragraph to discuss the LGM and 2xCO2 AMOC:**

*The climate sensitivity of the optimised parameter set is 3.2°C. The maximum Atlantic overturning is 17.8Sv, at a depth of 1.1km with the 10Sv contour, an indicator of the location of NADW formation, at a latitude of 56°N. Under LGM forcing, Atlantic overturning weakens to a peak of 11.1Sv at a depth of 1.0km and the 10Sv contour shifts southward to 45°N. Under doubled $CO_2$ forcing, Atlantic overturning weakens substantially to a peak of 7.6Sv at a depth of 0.4km.*

**The Atlantic meridional streamfunctions are shown below for your interest - we have not included these plots in the revised manuscript, but would be happy to do so if the reviewers or editor would like.**

[Figure]

- the strategy used in Araya-Melo et al. 2015 of summarising ice sheet forcing with a single quantity, and which is applied here by Holden et al., can be problematic. There is nothing to guarantee that Weschelian ice sheets were located similarly to those of the Early Pliocene, and also, as the authors rightly acknowledged, the build-up and decay phases of ice sheets are quite asymmetrical during the late Pleistocene. This might not be that much of a problem in certain applications, but it can be heavily misleading to users who would use this product in Europe or in Siberia without much discernment. For example, I would be particularly worried of archaeologists using the provided reconstructions near the limits of ice margins. A 3-D reconstruction of the Pleistocene can be very popular, so it needs to be disseminated wisely.

**We agree and have now included an extensive Section 9 detailing the assumptions and weaknesses of the approach (see below)**

This little experience brings me to the following, and related comments, about the present contribution by Holden et al.

1. It definitely needs to come up with appropriate health warnings about usage limits of the reconstruction. This is particularly crucial since the introduction presents it

**This is a very good point, and we have now included a section detailing the assumptions and weaknesses of the approach. We feel strongly that this approach should be published for dissemination, and would welcome alternative model variants (such as derived from LOVECLIM) to better quantify modelling uncertainties. Ecologists have great need of this data and they have long used (or "misused") whatever paleo-climate estimates are available. As you note yourself, such data are in very high demand. However, spatially-explicit time series of models are rarely available, usually only a few time-slices, so that a common practice is to apply linear interpolations in space and time. Our estimates are therefore a substantial improvement from the best available existing approaches (at least prior to 140ka). Macroecologists are interested in broad-scale spatiotemporal patterns, usually using >1x1 degree cells, and >1Ky time interval and, in general, the dynamics matter much more to them than the exact temperature/precipitation values. We expect that ecologists will understand very well that our paleo-climate data are estimates, following the general principle of modelling (all are wrong, but some are useful), although we certainly agree with the reviewer that detailing the weaknesses of the approach will be of substantial benefit as ecologists may not appreciate the specifics, and have added the new Section 9:**

*9 Limitations of the approach*

[revised manuscript text omitted]

2. a product oriented to end-users including archaeologists and biodiversity experts. A critical evaluation of the validity of the reconstructions near the North Atlantic (with emphasis on ocean circulation effects) needs to be provided.

**See new section 9 above**

In the introduction it is clearly said that uncertainty attached to the emulator (here, as a surrogate of GENIE-PLASIM) is distinct from the model uncertainty. This is true, and hence what would think that the evaluation (or "validation") of the emulator (as a surrogate) should be clearly distinguished from the evaluation of the model as a representation of the real climate. I found that this important distinction is pretty blurred in the section 6 (strangely divided into a section heading an a subsection 6.1). In fact there is very little about the evaluation of the emulator as a surrogate of GENIE-PLASIM. The authors refer to the PMIP ensemble and feel comfortable that the emulator-based reconstruction is in broad agreement with PMIP simulation of the mid-Holocene and the LGM, but in doing this the authors are mainly evaluating the reconstruction, not the emulator as a statistical surrogate. And, as I suggest in point 1. above, this evaluation is not providing with non-climatologist users with enough information about its application domain (the dos and don'ts). It is also quite uncomfortable that the emulator provides so- called bioclimatic variables (MIN and MAX over the seasonal cycle) while the "validation" is made on the basis of seasonal averages.

**Our main interest here is to derive useful reconstructions for paleo-applications, and so we regard a comparison of the emulated outputs with existing reconstructions as the most important test, capturing simultaneously the climate model errors and the emulation errors. We think that the faithfulness of the emulator wrt the simulator will be of less direct interest to ecologists and other users, but have expanded the cross-validation section in order to better quantify sources of emulation error and have added analysis to cross-validate the seasonal and annual average emulators used in the comparisons with multi-model intercomparisons:**

*Table 3 summarises the cross-validation of the eight emulators (i.e. four bioclimatic variables, two forcing categories). The second column tabulates the percentage of variance explained by the leading ten principal components, $\sum_{c=1,10} V_c$, and represents the maximum variance that could be explained by the emulators if they were perfect. The remaining columns tabulate the metric P when building the emulator with a series of different covariance functions, being the alternatives available in the DiceKriging R package (Roustant et al 2012). The reduction in variance explained (relative to column 2) reflects additional errors due to emulation.*

*The temperature decompositions explain 94-99% of the ensemble variance, compared to 87-90% for the precipitation decompositions. Under emulation, the variance explained is 81-98% for the temperature fields and 73-83% for precipitation fields. The emulator performance is weaker for precipitation, because the low order components needed to explain much of the ensemble variability are more difficult to emulate.*

*The power exponential was found to give comparable or better performance compared to the other covariance functions in all eight emulators and was therefore chosen as the default covariance function, and used in all analysis that follows.*

*Table 4 summarises the variance explained under cross-validation of the seasonal and annual average emulators used in the following Section 7. DJF (JJA) temperature emulator performance is similar to Min (Max) temperature emulator performance, suggesting that northern hemisphere temperature is more difficult to emulate than southern hemisphere temperature, as would be expected for the ice-sheet emulator in particular. The performance of the various seasonal precipitation emulators is similar (82.7% to 84.8% for the orbit and $CO_2$ emulator, 72.4% to 75.4% for the ice-sheet emulator), but annual precipitation is easier to emulate than seasonal precipitation (88.6% for the orbit and $CO_2$ emulator, 81.9% for the ice-sheet emulator).*

|  | DJF | JJA | Max | Min | Mean |
|---|---|---|---|---|---|
| *Orbit and $CO_2$ emulator* | | | | | |
| *Precipitation* | *84.8%* | *83.9%* | *82.7%* | *82.7%* | *88.6%* |
| *SAT* | *95.0%* | *97.8%* | *98.1%* | *95.0%* | *96.7%* |
| *Ice-sheet emulator* | | | | | |
| *Precipitation* | *74.0%* | *72.4%* | *75.4%* | *73.3%* | *81.9%* |
| *SAT* | *82.1%* | *94.8%* | *95.1%* | *80.9%* | *90.4%* |

*Table 4. Seasonal and annual mean emulator performance (as used in Section 7), measured by the metric P (Eq. 3, including ten components). A power exponential covariance is used in all cases. Note that max and min values repeat data from Table 3.*

3. Downscaling. Is this correct that downscaling as presented here assumes con- stant sub-grid correction anomaly, defined as the difference between the present- day observations and simulated grid-box-mean in a reference experiment (the anomaly being on the log of precipitations in the low-precipitation areas)? This treatment is arguably inadequate in palaeoclimate applications, where topogra- phy, surface type (think of Swiss glaciers to take

but one example), ice sheet mar- gins, land-sea mask, and ecotone boundaries vary substantially. Again, aren't we misleading the users by providing the illusion of a high-res reconstruction, while it may in fact be quite wrong at places? For reference, Levavasseur and co-authors have provided some thoughtful contributions about downscaling in palaeoclimate applications (e.g.: Levavasseur et al., 2010, The Cryosphere, 10.5194/tcd-4-2233-2010, and subsequent references).

**Yes, this is the approach we have taken, following Osborn et al 2016. We agree that there are limitations applying this approach to paleoclimate. These limitations were largely the reason for our choice of timeframe, as the assumptions would become increasingly untenable in deeper time, for instance as tectonic uplift progressively impacts on the validity. We have included a discussion of the weaknesses of the approach in the new section 9 (above)**

Minor comments
L. 280-295: Define the R-squared score

**Text added:**
*where $R_c^2$ is the coefficient of determination of the emulator of principal component c, evaluated under leave-one-out cross-validation of all simulations*

clarify what is mean by "the performance averaged over the eight emulators" (since the P-metric is near one, an arithmetic mean of P may be inadequate).

**The arithmetic mean is not required and text revised to:**
*The power exponential was found to give comparable or better performance compared to the other covariance functions in all eight emulators and was therefore chosen as the default covariance function, and used in all analysis that follows.*

Clarify the approach implemented for calibrating the length-scales appearing in covariance functions: do they vary across components, across variables?

**Text added:**
*We used an anisotropic covariance function (different length scales for each input dimension) and estimated the unknown length scale parameters using the type II maximum likelihood estimators (Rasmussen and Williams, 2006).*

The original reference for Berger and Loutre, as quoted here, is Berger and Loutre (1991), doi 10.1016/0277-3791(91)90033-Q . It could be cited along with the Pangea reference.

**Added**

Figures 3 and 4 are a bit overwhelming, with small character size, and only the one with eyes trained in deciphering PMIP-type experiments will understand that the anomalies seen here are reasonably expected from GENIE-PLASIM and understand its limits.

**We have removed the 2-component analysis to simplify the presentation, and have increased font sizes for improved clarity.**

0.2 Conclusion
The article could be a nice addition to current efforts in simulating the Pleistocene climate, but it is ambiguous as to its objective. If the authors ambition is to provide an technical, significant improvement on emulation, then they need to focus more on the evaluation of the emulator as such, and be more thorough in the discussion of the different technical options. If the ambition is to provide a final product to be used by non-climate users, then I would urge the authors to be much more critical about the pitfalls of the current reconstruction, and in the present state, I would actually discourage dissemination of this product, since the risks of it being misused are too large.

**We hope that the new Section 9 has addressed this important concern.**

---

## Author Comment (AC2) · 18 Jun 2019

**We thank the reviewer for this thorough and useful review. Our responses are in bold face and the associated revisions to the manuscript are in italics**

First, I apologize to the authors for taking so long to complete this review of "PALEO- PGEM v1.0: A statistical emulator of Pliocene-Pleistocene climate" for publication in GMD. In this paper, the authors describe the building of a statistical emulator from the PLISM-GENIE intermediate complexity model in order to provide a continuous recon- struction of important climate variables over the last 5 millions years, using interpolation to downscale these variables to a higher resolution. This paper logically follows a number of previous articles describing both building of emulators of intermediate complexity models (e.g., Holden et al. 2014) and their application to a range of important scientific questions, such as the links between climate and biodiversity (e.g. Range et al. 2018). It adds in particular a representation of ocean dynamics, which was lacking in previous studies, via the use of PLASIM-GENIE, the coupling of the two models having been described previously (Holden et al. 2018). Overall, the paper is well written and the methodology is sound. I thus recommend publication provided that some important points are taken care of or better discussed:

1. There is no evaluation of the impact of the addition of ocean dynamics on the climate reconstructions, despite being one of the main justifications of the paper (l. 48-49, l. 430). How different would be the results using PLASIM-ENTS rather than PLASIM-GENIE? This leads me to another question: in the PLASIM-GENIE experi- ments with varying ice sheets, is the land/sea mask changed accordingly or are only the height/albedo feedbacks taken into account (with no varying coastlines impacting ocean dynamics)? If the land/sea mask is modified, I wonder how valid are the transition between ocean modes linked to varying sea level in the subsequent emulations of the climate?

   **We have added a paragraph to discuss the LGM and 2xCO2 AMOC:**
   *The climate sensitivity of the optimised parameter set is 3.2°C. The maximum Atlantic overturning is 17.8Sv, at a depth of 1.1km with the 10Sv contour, an indicator of the location of NADW formation, at a latitude of 56°N. Under LGM forcing, Atlantic overturning weakens to a peak of 11.1Sv at a depth of 1.0km and the 10Sv contour shifts southward to 45°N. Under doubled $CO_2$ forcing, Atlantic overturning weakens substantially to a peak of 7.6Sv at a depth of 0.4km.*
   **The Atlantic meridional streamfunctions are shown below for your interest - we have not included these plots in the revised manuscript, but would be happy to do so if the reviewers or editor would like.**

[Figure]

**We cannot meaningfully compare PLASIM-GENIE with PLASIM-ENTS as the two models do not share the same tuned parameter values. However, the results will clearly be affected, for instance by the ocean circulation changes discussed above. While the effects of ocean dynamics on the climate reconstructions could be studied in general, along with an almost unlimited range of other questions regarding the climate dynamics themselves, this is outside the scope of the present paper which is focused on presenting the reconstructions as a tool for interdisciplinary studies.**

**The land-sea mask is held fixed at the present day. We have clarified this in the text and discussed some of the weaknesses associated with this in the new section 9 (see below).**

2. Related to the above, the anomaly method for the downscaling approach supposes that the biases between an observed climate state Ct and the emulated climate Et remain somehow constant through time. This is a very crude assumption given, among others, the documented changes in ocean circulation dynamics across the last 5 Ma. How well is the ocean circulation represented in the model? And ocean circulation changes, e.g. between the LGM and MH?

**See above re ocean circulation. We agree that the downscaling assumes the modelled bias remains fixed through time, and have included this in an extended discussion of weaknesses (section 9 reproduced below), which also discusses aspects of ocean circulation, in response to both reviewers.**

*9 Limitations of the approach*

[revised manuscript text omitted]

3. I am a bit disappointed by the lack of actual comparison to data over the last 5 Ma. For instance, how does it represent the mid-Pliocene period (and compare to the extensive database of PlioMiP exercises) or glacial-interglacial T and P variations? And transient events like MIS M2?

**We have added comparisons on glacial-interglacial variability (c.f. Koehler et al 2010), last interglacial transients (cf. Bakker et al 2013) and the Mid-Pliocene (c.f. Haywood et al 2013).**

*7.3 Glacial-interglacial variability*

[revised manuscript text omitted]

4. I think a concise but very general reminder of what is an emulator and of the theoretical basis behind could be useful for non specialized readers. In the present version, this reduces to one sentence (l. 58-59) because the Sections 3 and 4 either use too many reference to previous papers (Section 3) or are probably too specialized already (Section 4).

**We have extended this as below:**
*Our methodology uses principal component analysis to project spatial fields of model output onto a lower dimensional space of the dominant simulated patterns of change and then derives regression relationships between the simulator inputs and the coefficients of the dominant patterns. The method is analogous to the widely-used pattern-scaling technique (Tebaldi and Arblaster 2014), which assumes that an invariant pattern of simulated change can be scaled by global warming. Our approach extends this by including several (here ten) principal components for each climate variable, thereby allowing us to capture nonlinear patterns of change. The regression approach we use involves Gaussian process (GP) emulation (Rasmussen 2004).*

5. The discussion on the limits of the approach and of the (numerous) uncertainties and approximations made should be expanded, in particular if the aim is to provide a widely available climatic reconstruction that is then used to force ecological niche modelling or biodiversity models (l. 36-42). Because this requires at least some confidence in the ability of the model to reproduce "true" (absolute) paleoclimatic conditions and variability (see points 1-3 above).

**See new section 9 (above)**

Other minor details l. 52 "naïve simulation would not be possible for an application of this ambition". I do not understand this sentence.

**We have revised this as below (full paragraph included for context)**

*However, simulation alone would not be possible for an application of this ambition. We use the computationally-fast low-resolution AOGCM PLASIM-GENIE (Holden et al 2016), but even with this relatively simple model a five million-year transient simulation would demand ~300 CPU years of computing, which could not readily be parallelised. We overcome this intractability by using statistical emulation.*

l. 272-274. "GPs are highly flexible non-parametric regression models which have greater modelling power than linear models". Please clarify.

**Clarified with**
*Linear models live in a finite dimensional space defined by polynomial functions of the covariates. Gaussian processes live in a much richer space of functions.*

l. 393. "Warm biases are more modest". On the basis of the figure, this is hard to believe as there are regions with a warm bias as large as the cold bias of other regions.

**Clarified with:**
*Warm biases are more modest except for the Tibetan Plateau and Andes where the lapse rate cooling in these narrow mountain chains is poorly resolved by the climate model (but corrected for by the downscaling approach described below).*

Fig. 3 and 4. Please use the same color scale and range as the PMIP ensemble to ease comparisons. Why is the Southern Ocean cooling rather than warming in the MH simulation?

**We have removed the 2-component analysis to simplify the presentation, and have increased font sizes for improved clarity. Unfortunately we were unable to access the PMIP2 netcdf files, so could not use the same colour scales. In general we use the same ranges in both plots, but in some cases we prefer to leave these different for adequate contrast in the legend scale. Most notably the SD of LGM temperature is only 5 degrees in PALEO-PGEM, compared to 20 degrees in PMIP2, reflecting the fact that we do not sample uncertain climate sensitivity. We have added a note in the caption to emphasise this.**

**Re emulated Southern Ocean cooling in the MH, we have added the text.**

*The most significant difference is Antarctic cooling of ~3°C in PALEO-PGEM, which contrasts with a warming signal in the ensemble mean of PMIP2 (although we note DJF Antarctic cooling of 0.5°C was simulated in HadCM3M2). A significant cold Antarctic bias is also apparent during the Last Interglacial (Section 7.4). High southern latitudes are poorly modelled by PLASIM-GENIE. The preindustrial state exhibits a warm Antarctic bias, with greatly understated sea ice, a slow Antarctic Circumpolar Current and weak, northerly shifted zonal winds (Holden et al 2016), which are likely to be associated with well-known*

*difficulties of resolving Southern Ocean wind stress at low meridional resolution (Tibaldi et al 1990, Schmittner et al 2010).*